**RESEARCH**                                          **Open Access**

# RNA editing in cancer impacts mRNA abundance in immune response pathways

Tracey W. Chan[1†], Ting Fu[2†], Jae Hoon Bahn[3], Hyun-Ik Jun[3], Jae-Hyung Lee[3,4], Giovanni Quinones-Valdez[5], Chonghui Cheng[6] and Xinshu Xiao[1,2,3,7,8,9*]

* Correspondence: gxxiao@ucla.edu
[†]Tracey Chan and Ting Fu contributed equally to this work.
[1]Bioinformatics Interdepartmental Program, UCLA, Los Angeles, CA, USA
[2]Molecular, Cellular and Integrative Physiology Interdepartmental Program, UCLA, Los Angeles, CA, USA
Full list of author information is available at the end of the article

## Abstract

**Background:** RNA editing generates modifications to the RNA sequences, thereby increasing protein diversity and shaping various layers of gene regulation. Recent studies have revealed global shifts in editing levels across many cancer types, as well as a few specific mechanisms implicating individual sites in tumorigenesis or metastasis. However, most tumor-associated sites, predominantly in noncoding regions, have unknown functional relevance.

**Results:** Here, we carry out integrative analysis of RNA editing profiles between epithelial and mesenchymal tumors, since epithelial-mesenchymal transition is a key paradigm for metastasis. We identify distinct editing patterns between epithelial and mesenchymal tumors in seven cancer types using TCGA data, an observation further supported by single-cell RNA sequencing data and ADAR perturbation experiments in cell culture. Through computational analyses and experimental validations, we show that differential editing sites between epithelial and mesenchymal phenotypes function by regulating mRNA abundance of their respective genes. Our analysis of RNA-binding proteins reveals ILF3 as a potential regulator of this process, supported by experimental validations. Consistent with the known roles of ILF3 in immune response, epithelial-mesenchymal differential editing sites are enriched in genes involved in immune and viral processes. The strongest target of editing-dependent ILF3 regulation is the transcript encoding PKR, a crucial player in immune and viral response.

**Conclusions:** Our study reports widespread differences in RNA editing between epithelial and mesenchymal tumors and a novel mechanism of editing-dependent regulation of mRNA abundance. It reveals the broad impact of RNA editing in cancer and its relevance to cancer-related immune pathways.

## Introduction

RNA editing, the modification of specific nucleotides in RNA sequences, expands diversity in proteins and gene regulatory mechanisms [1, 2]. The most frequent type of RNA editing in human cells is A-to-I editing, which refers to the deamination of adenosine (A) to inosine (I) and is catalyzed by the adenosine deaminases acting on RNA (ADAR) family of enzymes [3]. Three ADAR genes are encoded in the human genome,

namely ADAR1, ADAR2, and ADAR3. Catalytically active ADAR1 and ADAR2 are widely expressed across tissues. In contrast, ADAR3 is exclusively expressed in certain brain regions and is catalytically inactive [4]. As inosine is recognized as guanosine (G) in translation and sequencing, A-to-I editing is also referred to as A-to-G editing. Though millions of editing events have been revealed across the human transcriptome, only a small proportion of editing events have been functionally characterized. The effects of most editing sites, primarily within noncoding regions, have yet to be discovered [5, 6].

Increasing evidence has established the importance of RNA editing dysregulation in cancer. A number of studies have delineated mechanisms through which individual RNA editing sites, mostly causing recoding events (i.e., amino acid changes), promote or suppress tumor development [7–10]. Besides functioning in tumorigenesis, edited RNA transcripts can be translated into edited peptides, which may be recognized as cancer antigens and activate an anti-tumor immune response [11, 12]. Furthermore, across various cancer types, genome-wide aberrations in RNA editing were observed and associated with clinical features [13–15]. Within each cancer type, editing levels generally increased or decreased in tumors, compared to matched normal samples. Editing levels of specific sites were correlated with tumor stage, subtype, and patient survival, and for a smaller subset of nonsynonymous sites, editing altered cell proliferation and drug sensitivity in cell line experiments [13]. As RNA editing has the potential to inform development of improved cancer diagnostics and patient-specific treatments, thorough investigation of the precise functions and regulatory mechanisms of the many cancer-type-specific RNA editing changes is needed [10].

In cancer progression, activation of epithelial-mesenchymal transition (EMT) facilitates metastasis by enabling tumor cells to gain an invasive phenotype, infiltrate the circulatory and lymphatic systems, and reach distant sites for colonization [16–18]. A few RNA editing sites have been associated with this process so far. Specifically, editing events in SLC22A3, FAK, COPA, RHOQ, and miR-200b were demonstrated to accelerate metastasis [12, 19–23]. While miR-200b normally targets ZEB1 and ZEB2, which are key EMT-inducing transcription factors, editing alters its targets and enhances cell invasiveness and motility [23]. The SLC22A3 recoding event also promoted EMT, causing expression changes in EMT marker genes [19]. In contrast, a recoding event in GABRA3 inhibited metastasis and was present only in non-invasive cell lines and non-metastatic tumors [22]. These studies highlight the functional relevance of RNA editing in metastasis and the merit of a more comprehensive investigation.

Here, we present a global analysis and comparison of RNA editing profiles between epithelial (E) and mesenchymal (M) phenotypes of primary tumors across multiple cancer types. Using RNA-seq data derived from bulk tumors and single cells, we observed distinct editing patterns between phenotypes, with editing differences often enriched among immune response pathway genes. Supported by experimental evidence, we show that differential editing sites affect host gene mRNA abundance and identify a novel mechanism of editing-dependent stabilization of mRNAs by ILF3. One of the target genes of ILF3 is EIF2AK2, coding for protein kinase R (PKR), a key player in immune and viral response.

## Results
### Altered RNA editing profiles between epithelial and mesenchymal tumors
EMT is known to be accompanied by substantial transcriptome remodeling [17, 24–28]. Given the previously reported functional relevance of RNA editing in EMT [19, 23, 29],

we hypothesized that epithelial and mesenchymal tumors possess different transcriptome-wide RNA editing profiles. Thus, we analyzed RNA-seq datasets of primary tumors from The Cancer Genome Atlas (TCGA). We focused on seven cancer types that have been previously studied in the context of EMT and have relatively large sample sizes available from TCGA (Fig. 1a). To classify tumors into epithelial (E) and mesenchymal (M) phenotypes, we utilized a well-established EMT scoring system, where scoring and categorization of tumors into these E and M phenotypes enabled systematic identification of cancer-specific differences in treatment response between phenotypes, as well as associations with survival [30]. Of all categorized tumors for each cancer type, we further refined the subset of tumors such that metadata were matched between the two groups (Additional file 1: Table S1).

Applying our previously published methods [1, 31, 32], we quantified editing levels at over 4 million editing sites recorded in the REDIportal database [33]. We then identified sites that were differentially edited between E and M tumors in each cancer type. To control for false discoveries, we filtered out predicted differential editing sites that repeatedly exhibited differences in editing when phenotype labels were shuffled randomly. Principal component analysis on differential editing levels showed that E and M tumors could be well separated by the first two principal components of editing

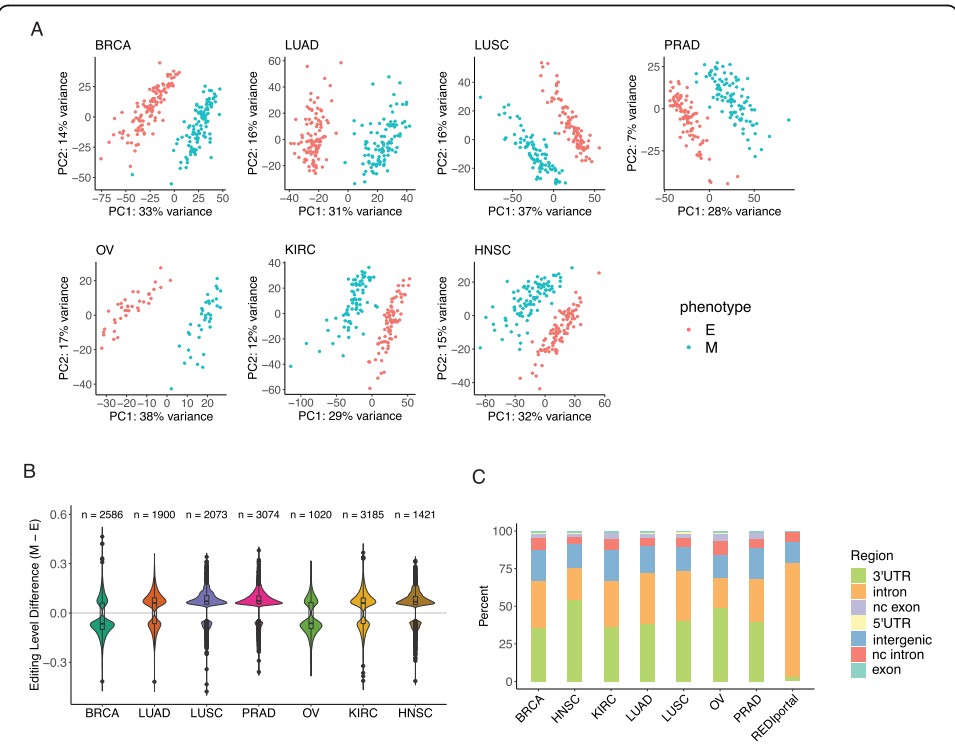

**Fig. 1** Overview of differential editing in cancer EMT. The following cancer types were studied: breast invasive carcinoma (BRCA), lung adenocarcinoma (LUAD), lung squamous cell carcinoma (LUSC), prostate adenocarcinoma (PRAD), ovarian serous cystadenocarcinoma (OV), kidney renal clear cell carcinoma (KIRC), head and neck squamous cell carcinoma (HNSC). **a** First two principal components of differential editing profiles separate tumor samples into epithelial (E) and mesenchymal (M) phenotypes across cancer types. **b** Distributions of differences in mean editing levels between E and M tumors in each cancer type. The number of differential editing sites is listed on top of each distribution. **c** Differential editing sites are mostly found in 3′ UTR and intronic regions in all cancer types, with higher proportions of 3′ UTR sites compared to that of all editing sites from the REDIportal database

(Fig. 1a). These first two principal components did not appear to be confounded by sample metadata and suggest that most of the variation in editing is explained by the distinction of E and M phenotypes (Additional file 2: Fig. S1).

Based on the differential editing sites, most cancer types, including LUAD, LUSC, PRAD, KIRC, and HNSC, demonstrated a hyperediting trend in the M phenotype (Fig. 1b). In contrast, two cancer types, BRCA and OV, had a trend of hypoediting in the M samples. The majority of differential editing sites in all cancer types were located in the 3′ untranslated regions (UTRs) or introns (Fig. 1c). The above results suggest that distinct RNA editing profiles exist between E and M phenotypes.

### Editing patterns are shared among cancer types and distinct from differential expression

Given dominant trends of hyperediting or hypoediting that distinguished E and M phenotypes in an individual cancer type, we asked whether genes with differential editing patterns were shared or distinct across cancer types. We examined the statistical significance of overlap in differentially edited genes between pairs of cancer types by Rank-rank Hypergeometric Overlap (RRHO). Extending Gene Set Enrichment Analysis (GSEA) to two dimensions, RRHO tests the significance of the intersection of gene lists, ranked by a metric of differential expression, across two genome-wide datasets [34]. We applied RRHO to RNA editing here by ranking genes according to the significance of tested editing differences between E and M and the direction of editing differences ("Methods"). In addition to shared directionality of global editing trends, we found significant overlap in genes with editing changes among multiple cancer types (Fig. 2a). Within pairs of cancer types, most significant overlaps were enriched at the bottom left or top right corners, where genes were hyperedited or hypoedited in both cancer types, respectively. These significant overlaps in genes based on differential editing suggest that editing changes in EMT may affect common pathways across cancer types.

It should be noted that differentially edited genes do not overlap with differentially expressed genes (Fig. 2b). This observation indicates that gene expression changes in EMT did not confound the RNA editing differences observed. Thus, altered editing potentially represents a distinct layer of molecular changes in EMT.

### Differential editing occurs in genes of immune relevance

Next, we examined the gene ontologies enriched among genes with differential editing in EMT. In this analysis, background control genes were chosen randomly from those that did not have differential editing sites but had similar gene length and GC content as the differentially edited genes ("Methods"). Across multiple cancer types, differentially edited genes were enriched with viral-host interaction features, interferon (IFN), and other immune response pathways, metabolic processes, and translational regulation (Fig. 2c, Additional file 2: Fig. S2).

The observation of immune-relevant categories is of particular interest. RNA editing has been described as a mechanism to label endogenous double-stranded RNAs and consequently prevent IFN induction [35–39]. However, the roles of editing events in genes directly associated with immune response, such as those in the IFN response pathways, have not been well characterized. Our observation indicates that RNA editing may directly affect immune response genes in EMT.

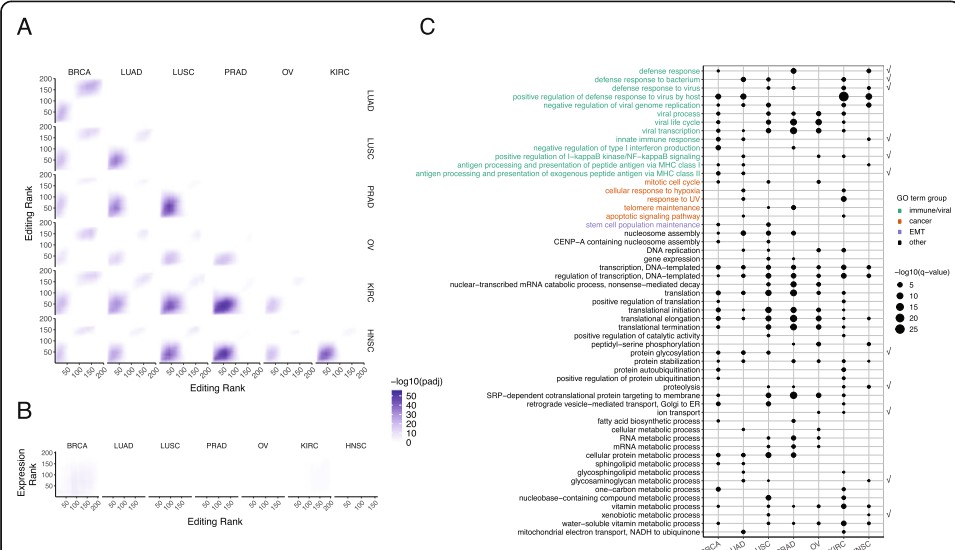

**Fig. 2** Differential editing patterns are shared among cancer types yet distinct from differential gene expression. **a** Rank-rank hypergeometric overlap (RRHO) map of RNA editing across pairs of cancer types. Each heatmap (for two cancer types) represents the matrix of log10-transformed adjusted *p* values that indicate the extent of overlap in two gene lists at each possible pair of ranks. For an individual cancer type, genes were ranked by the signed significance of RNA editing differences (M-E). Genes with higher editing in the M phenotype are at lower ranks, while those with higher editing levels in E tumors are at higher ranks. Higher pixel darkness indicates stronger enrichment of overlapping genes within the rank thresholds given by the *x* and *y* coordinates. The step size between ranks was 30 genes. **b** RRHO map of editing and gene expression within each cancer type. Each heatmap contains log10-transformed adjusted *p* values of hypergeometric overlap between genes ranked by editing differences (*x*-axis) and genes ranked by expression differences (*y*-axis) in a single cancer type. Similar to ranking genes by differential editing, genes were ranked by the signed significance of expression differences, such that genes at lower ranks have higher expression in M tumors, while genes at higher ranks have higher expression in the E phenotype. The step size between ranks was 30 genes. **c** Significance of enrichment of gene ontology (GO) terms in differentially edited genes of each cancer type represented by point size (log10-transformed adjusted *p* value). Terms significantly enriched in at least two cancer types are shown. Check mark on the right indicates terms that were also significantly enriched in differentially expressed genes in at least two cancer types. Text color indicates category of biological relevance

## Contribution of cell types to differential editing

Given the observed enrichment of differential editing in immune-relevant genes, we asked whether our identified differential editing events primarily occur in cancer cells or in other cell types in the tumor microenvironment. To address this question, we analyzed single-cell (sc) RNA-seq data from three non-small cell lung cancer (NSCLC) patients, each with three tumor samples from the tumor edge, core, and in-between [40]. Following quality control measures, we clustered the cells in two rounds and labeled cell types based on marker genes to obtain T cells, B cells, myeloid cells, endothelial cells (EC), fibroblasts (Fibro), epithelial cells (Epi), mast cells, alveolar cells, and cancer cells (Additional file 2: Fig. S3A-C, "Methods"). Supporting the accuracy of this clustering, expression of marker genes was generally highest in their expected cell types when RPKM was calculated from pooled cells and when a signature gene expression matrix was predicted by CIBERSORTx [41] (Additional file 2: Fig. S3D).

To gauge the contribution of individual cell types to bulk tumor differential editing, we examined gene expression and editing profiles of each cell type. Specifically, we pooled cells of each type and calculated the percent of differentially edited genes from the bulk tumor analysis that were expressed in each cell type. Cancer cells expressed the highest

proportion of genes that were differentially edited (Fig. 3a). We then measured the extent of editing in each cell type by calculating the percent of bulk tumor differential editing sites that were edited. Consistent with the expression analysis, the highest proportion of differential sites was edited in cancer cells (Fig. 3b). Therefore, the editing differences observed among bulk tumors may be mainly attributable to the cancer cells.

We next separated cancer cells to epithelial and mesenchymal cell clusters (Fig. 3c, "Methods"). Sampling epithelial cells to match mesenchymal cells in terms of cell number (200 cells) and metadata, we pooled cells within each phenotype together and detected RNA editing events (Additional file 2: Fig. S4). Although the scRNA-seq primarily sequences the 3′ ends of mRNAs, a relatively small number of RNA editing events were still captured. We identified nine editing sites with significant differences between E and M (Fig. 3d). All nine differential sites exhibited higher editing levels in the M phenotype, which is consistent with the hyperediting trend in M observed in bulk LUAD and LUSC tumors (Fig. 1b). Two sites overlapped with differentially edited sites in LUAD or LUSC and both had hyperediting in M cells, consistent with the direction in bulk tumors (Additional file 2: Fig. S5). This small overlap likely reflects the low coverage on editing sites in the single-cell data, and/or the possibility that more differential editing sites, which were not identified in our study due to limits in power, exist in the bulk tumors.

Notable differentially edited genes include RHOA, which is active in cell migration and is associated with metastasis in multiple cancer types [42–44], and ARL16, a reported negative regulator of RIG-I activity [45], consistent with the observed enrichment of immune-relevant genes that were differentially edited in bulk tumors. Overall, the findings from single-cell data support the hypothesis that editing differences between bulk E and M tumors mainly reflect changes occurring in cancer cells.

### ADAR1 or ADAR2 knockdown induced EMT

Given the differential editing profiles between E and M tumors, an important question is whether the editing changes are functionally relevant to EMT. To address this question, we first examined if changes in ADAR expression affect EMT. Using cell culture systems commonly employed in EMT studies, we carried out knockdown (KD) experiments of ADAR1 or ADAR2 in two cell lines, A549 and MCF10A, via siRNAs. Upon ADAR1 KD, A549 cells showed elongated spindle-like mesenchymal morphology (Fig. 4a). We also confirmed the loss of epithelial markers (E-cadherin and γ-Catenin) and gain of mesenchymal marker (Vimentin) in ADAR1 KD A549 cells (Fig. 4b). Similar results were observed upon ADAR2 KD in A549 cells (Fig. 4c, d) and reproducible in MCF10A cells (Fig. 4e, f). These findings suggest that loss of either catalytically active ADAR enabled EMT in the two cell lines. The phenotypic changes following ADAR2 KD are consistent with a previous report that ADAR2 deficiency can induce EMT in SW480 cells [29]. Together, these results indicate that knockdown of ADARs promotes EMT.

As expected, ADAR KD induced significant editing changes measured by RNA-seq in A549 cells (Additional file 2: Fig. S6A-B), with ADAR1 KD affecting a large number of editing sites but ADAR2 having fewer targets. A minority of ADAR2-

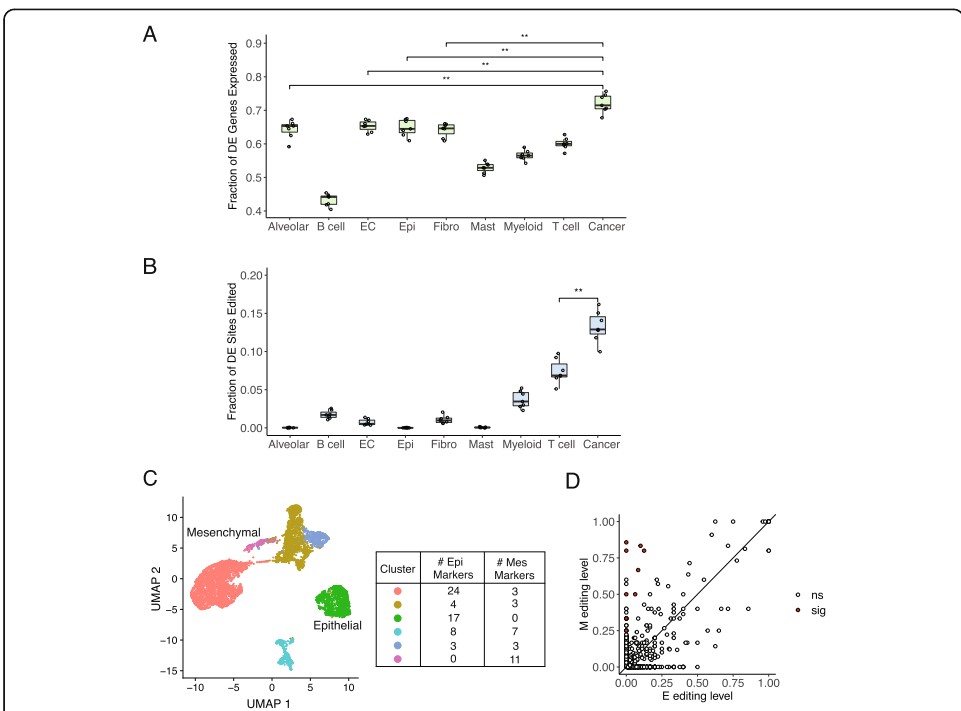

**Fig. 3** Contribution of cell types to differential editing. **a** Proportions of differentially edited (DE) genes from bulk tumor analysis that were expressed in cell types identified in lung cancer single-cell RNA-seq data. Each point represents the proportion of genes from one cancer type. A gene was considered as expressed in a cell type if its expression ≥ 1 RPKM. RPKM values were calculated within each cell type by pooling reads of the same cell type together. Proportions were compared for top cell types by Mann Whitney *U* test, with significance of *p* values shown. **\*\****p* ≤ 0.01. EC stands for endothelial cells. **b** Proportion of differential editing sites from bulk tumor analysis that were edited in individual cell types. A site was considered as edited in a cell type if the site was covered by at least 5 reads and editing was supported by at least 2 reads. Each point represents the proportion of sites from one cancer type. Proportions for top cell types were compared by Mann Whitney U test, with *p* value significance shown. **\*\****p* ≤ 0.01. **c** UMAP projection of 6526 tumor cells based on expression profiles, colored by cluster assignment (scatterplot, left). By differential expression of epithelial or mesenchymal markers (table, right), green and purple clusters were labeled as epithelial and mesenchymal, respectively. **d** Scatterplot of editing levels of pooled E and M cells, with *y* = *x* line. Editing sites exhibiting significant differences between E and M were labeled in red. Differences were considered significant if the difference between editing levels ≥ 0.05 and Fisher's exact *p* value < 0.05

responding sites had increased editing upon ADAR2 KD, reflecting the likely compensation by ADAR1. The reverse, compensation of ADAR1 loss by ADAR2, was not observed. Among the lung cancer E-M differential editing sites that were testable in the above A549 RNA-seq data, the vast majority responded to KD of either ADAR or double KD (Additional file 2: Fig. S6C). These results confirm the impairment of RNA editing at genome scale upon the loss of ADARs.

We next examined mRNA expression of ADARs in the bulk E and M tumors across cancer types. In several cancer types with a hyperediting trend in M, higher mRNA expression of ADAR1 or ADAR2 likely contributed to increased editing levels in M tumors (Additional file 2: Fig. S7). However, ADAR expression was not consistent with RNA editing differences for some cancer types. Thus, although ADAR KD caused EMT in cell culture models, ADAR expression alone may not sufficiently explain the global editing trends observed in bulk tumors.

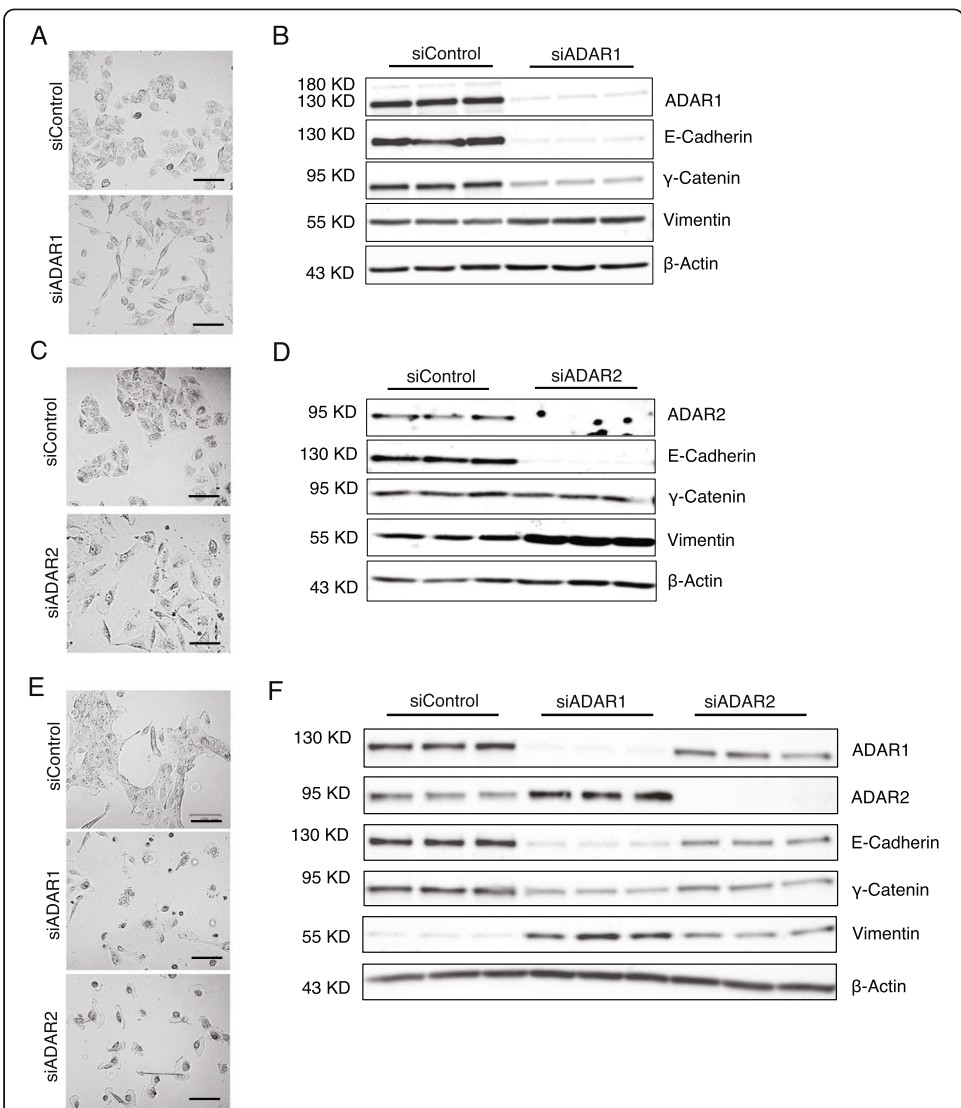

**Fig. 4** ADAR1 or ADAR2 knockdown induced EMT. **a** Images of A549 cells transfected with siRNAs for ADAR1 knockdown (KD) (siADAR1) or control siRNAs (siControl). Scale bars, 100 μm. **b** Loss of epithelial markers (E-cadherin and γ-Catenin) and induction of mesenchymal marker (Vimentin) in A549 cells upon ADAR1 KD. Cells were treated with 100 nM siRNA for 72 h. Three biological replicates were used in each condition. **c** Images of A549 cells transfected with siRNAs for ADAR2 KD (siADAR2) or control siRNAs (siControl). Scale bars, 100 μm. **d** Loss of epithelial markers (E-cadherin and γ-Catenin) and induction of mesenchymal marker (Vimentin) in A549 cells upon ADAR2 KD. Cells were treated with 11 nM siRNA for 72 h. Three biological replicates were used in each condition. **e** Images of MCF10A cells with ADAR1 or ADAR2 KD or control siRNAs. Scale bars, 100 μm. **f** Loss of epithelial markers (E-cadherin and γ-Catenin) and induction of mesenchymal markers (Vimentin) in MCF10A cells upon ADAR1 KD or ADAR2 KD. Cells were treated with 11 nM siRNA for 72 h. Three biological replicates were used in each condition

## Impact of RNA editing on mRNA abundance

Given ADAR's primary role in RNA editing, we next asked how RNA editing may affect genes relevant to EMT, especially those related to immune response (Fig. 2c). Since a relatively large fraction of differential editing sites is located in 3′ UTRs, we examined the hypothesis that these sites may affect mRNA abundance of their respective genes. Thus, we first calculated the correlation between editing levels and mRNA abundance for differentially edited sites observed in the E-M comparison. Using a regression model

accounting for confounding factors including age, gender and race, we observed a total of 127 genes whose editing sites are significantly correlated with mRNA abundance (FDR < 10%) in at least one type of cancer (Fig. 5a). In addition, among these genes, 77% (94 of 122 testable genes) demonstrated a significant correlation in at least one human tissue type based on a similar analysis of GTEx data, 78% (73/94 genes) of which showed the same direction of correlation between cancer and at least one GTEx tissue.

To further evaluate the regulatory role of RNA editing on mRNA abundance, we next examined the change in mRNA expression levels upon ADAR1 KD. We used ADAR1 KD RNA-Seq data from 5 cell lines: U87, HepG2, K562, HeLa, and B cells [1, 46, 47], respectively. Out of the 127 edited genes identified above, 126 of them were detectable at an expression level of at least 1 FPKM (and edited) in at least one cell line (control or ADAR1 KD condition). Among them, 71% (89 genes, red dots, Fig. 5b) showed inverse correlation between ADAR1 KD and editing level coefficient in at least one cell line (Fig. 5b). These genes showed an enrichment of negative expression changes upon ADAR1 KD, indicating a likely stabilizing effect imposed by RNA editing ($p = 2.7e{-}4$, binomial test). Among expression-correlated editing sites in the 89 genes, 64% are located in 3′ UTRs, a percentage that is significantly higher than that of E-M differential editing sites in general ($p = 2.4e{-}4$, Fig. 5c). We thus refer to the 89 genes as putative target genes whose expression is modulated by RNA editing (Additional file 1: Table S2).

Next, we experimentally validated the regulation of mRNA abundance by six editing sites within three genes: RNF24, RHOA, and MRPS16. We used a minigene reporter with bi-directional promoters for mCherry and eYFP [48] and cloned edited and unedited versions of each editing site and its surrounding 3′ UTR region into the 3′ UTR of mCherry. Using expression of eYFP as an internal control, we compared mCherry expression between cells carrying the edited and unedited versions for each editing site. All six editing sites induced significant expression differences in the direction consistent with the editing-expression correlations observed in primary tumors (Fig. 5d, Additional file 1: Table S3). While positive editing associations were dominant among predicted target genes, there also exist negative associations between editing and expression levels. We tested one example of the latter category (RHOA).

### ILF3 as an editing-dependent regulator of mRNA abundance

Since mRNA stability is closely regulated by RNA-binding proteins (RBPs) [49–52], we next asked whether RBPs are involved in the modulation of mRNA abundance by RNA editing sites. To this end, we analyzed enhanced ultraviolet crosslinking and immunoprecipitation (eCLIP) datasets of 126 RBPs in two cell lines (HepG2 and K562) from ENCODE [46, 53]. We asked whether RBP binding signals are enriched significantly closer to editing sites in the 89 potential target genes than expected by chance. This analysis identified ILF3 as a top protein with significantly short distances to the editing sites in both cell lines (Additional file 2: Fig. S8A). To validate this finding and test this relationship in a different cell type, we performed eCLIP-seq of ILF3 in A549 cells. The same observation was made via this dataset (Fig. 6a). As observed in HepG2 and K562 cells, differential editing sites within predicted target genes were significantly closer to ILF3 binding regions in A549 cells than random gene-matched control sites. Furthermore, 75 (84%) of the 89 genes showed a significant correlation between their gene expression

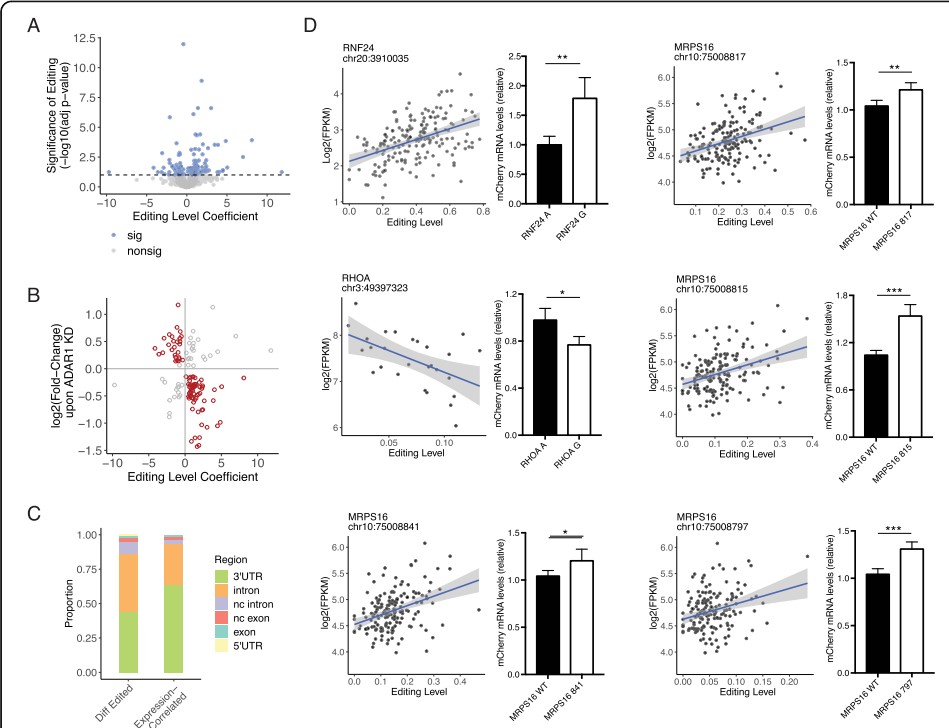

**Fig. 5** Effects of editing on mRNA abundance. **a** Scatterplot of coefficient estimate and statistical significance (log10-transformed adjusted *p* value) of editing level as a predictor of host mRNA expression in linear regression, accounting for potential confounding variables. For genes with multiple editing sites associated with expression, the most significantly associated site was used. Dashed line indicates significance threshold based on 10% false discovery rate (FDR). **b** Scatterplot of editing level coefficient estimate from multiple linear regression models used in A and log2-transformed fold change of the corresponding gene observed in ADAR1 KD cells. Red points indicate expression changes in the direction consistent with the sign of the editing association, in contrast to the gray points. **c** Editing sites associated with host expression (Expression-Correlated) are more often found in 3′ UTR regions, compared to all differential editing sites (Diff Edited, not including intergenic sites). **d** Validation of six editing sites affecting host mRNA abundance. For each site, a scatterplot of editing level and log2-transformed mRNA expression in the TCGA data is shown. On the right of each scatterplot is mCherry expression, normalized by eYFP expression, of minigenes with A or G, corresponding to nonedited or edited versions of the sites in the 3′ UTR of each gene. All minigenes were tested in Hela cells with five biological replicates. Normalized expression values (mean ± SD) were compared between edited and nonedited versions by two-sided *t*-test. \**p* < 0.05, \*\**p* < 0.01, \*\*\**p* < 0.001. Note that RHOA and MRPS16 editing sites were identified as differential sites in the single-cell RNA-seq analysis (Fig. 3c)

and the expression of ILF3 (FDR < 10%), 37 of which had an absolute correlation coefficient of at least 0.2 (Fig. 6b). Importantly, the majority of the significant correlations were positive, consistent with the known roles of ILF3 in stabilizing its target mRNAs [54–56].

## Impact of ILF3 on immune-relevant genes

ILF3 promotes an antiviral response through its binding to RNAs [57–59]. Given the fact that immune-relevant genes are differentially edited in E-M (Fig. 2c), we next asked whether ILF3 regulates the mRNA abundance of these EMT-associated differentially edited, immune-relevant genes. Among the 89 genes whose expression was affected by RNA editing, 20 genes fall into the immune or viral GO categories. Interestingly, the ILF3 binding sites were significantly closer to the differential editing sites of these 20

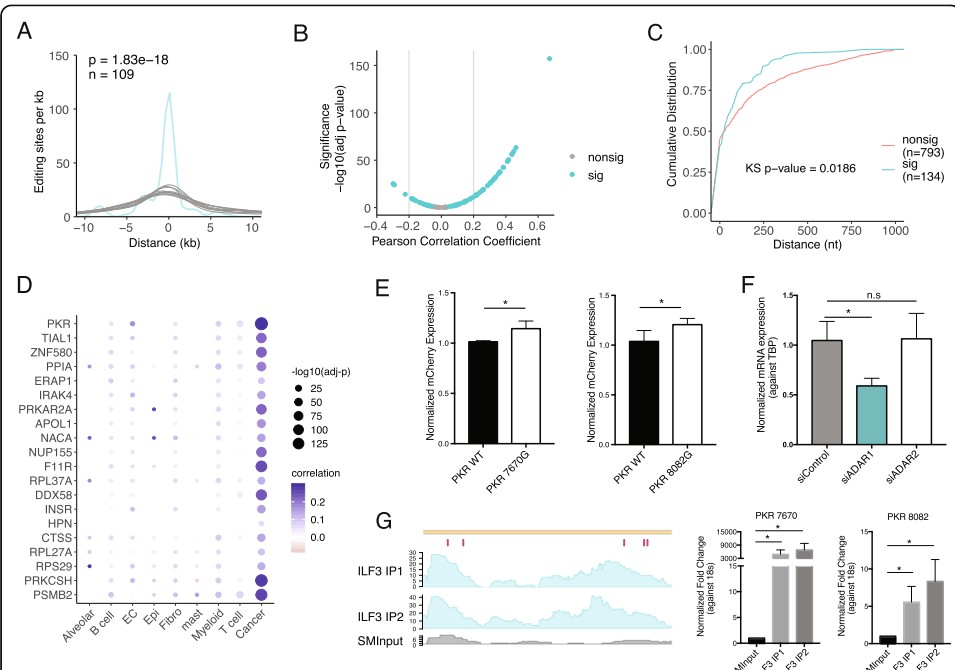

**Fig. 6** ILF3 binds closely to the differential editing sites in editing-expression-correlated genes. **a** Histogram of distances between differential editing sites in editing-correlated genes and the closest ILF3 eCLIP peaks in A549 cells (turquoise), up to 10 kb. Gray curves represent distances for 10 sets of randomly picked A's in the same genes as differential editing sites. Number of differential editing sites is given by *n*. *p* value was calculated by comparing the area under the curve (AUC) of the distance distribution for differential editing sites to a normal distribution fit to the AUC values of 10,000 sets of random gene-matched A's. **b** Scatterplot of Pearson correlation coefficient and significance (log10-transformed adjusted *p* value) of correlation between ILF3 mRNA expression and mRNA expression of editing-correlated genes. Genes passing 10% FDR are labeled as significant (sig, turquoise), others as nonsig. **c** Cumulative distributions of distances between ILF3 eCLIP peaks and differential editing sites within editing-expression-associated genes (sig) or differential editing sites in genes without editing-expression associations (nonsig), up to 1 kb. Only genes associated with immune and viral related GO terms were included. *p* value calculated by the Kolmogorov-Smirnov test. **d** For each cell type in the lung cancer scRNA-seq dataset, ILF3 mRNA expression was correlated with mRNA expression of editing-expression-correlated genes (identified in the TCGA data) by Pearson correlation. Genes associated with any immune or viral-related GO term are shown. The size of each point indicates significance of correlation and color corresponds to values of the correlation coefficient. **e** Normalized mCherry expression (mean ± SD) for nonedited or edited versions of sites in the 3′ UTR of PKR in A549 cells. Five biological replicates were performed. *p* value calculated by two-sided *t*-test (same below), **p* < 0.05. **f** Normalized mRNA expression (mean ± SD) of endogenous PKR in siControl, siADAR1, and siADAR2 A549 cells. Three biological replicates were performed. **p* < 0.05. n.s., not significant. **g** Read coverage of ILF3 eCLIP-seq in A549 cells for two biological replicates (ILF3 IP1 and ILF3 IP2, turquoise) and size-matched input (SMInput, gray). The five validated 3′ UTR editing sites affecting PKR mRNA abundance in A549 cells are labeled in magenta (left). Right: Validation of PKR eCLIP signal overlapping two editing sites. PKR expression (mean ± SD) was measured by qRT-PCR in the IP or SMInput samples and normalized against the expression of 18s rRNA, **p* < 0.05. (*n* = 3)

genes than differential sites in immune-related genes without editing-expression associations (Fig. 6c). Together, these results suggest that ILF3 binds close to the editing sites of immune-related genes.

Since we observed that differential editing between bulk E and M tumors mainly reflected changes occurring in cancer cells (Fig. 3a, b), we next asked whether the above regulatory relationship between ILF3 and immune-related genes also occurs in cancer cells. To this end, we analyzed gene expression of individual cell types identified in the NSCLC scRNA-seq dataset. Within each cell type, we correlated ILF3 expression with

expression of the 20 immune-related target genes. In cancer cells, all 20 genes had expression levels positively correlated with ILF3 expression at 10% FDR (Fig. 6d). Though significant correlations were also observed in other cell types, only cancer cells showed correlation coefficients of at least 0.2 in magnitude. This result suggests that the mRNA stabilizing function of ILF3 is prominent in cancer cells, in line with our observation that E-M differential editing primarily occurs in cancer cells.

### PKR expression is affected by 3′ UTR editing through ILF3 regulation

Among the 20 immune-related genes putatively regulated by ILF3, the gene EIF2AK2, coding for protein kinase R (PKR), had the most significant expression-editing correlation (Additional file 1: Table S2) and expression correlation with ILF3 (Fig. 6d). Activated by dsRNA, PKR suppresses translation and promotes apoptosis through its phosphorylation activity [60, 61]. PKR also regulates various signaling pathways, such as NF-κB and p38 MAPK, in response to cellular stress [60]. Using the editing minigene reporter, we examined the individual effects of seven 3′ UTR editing sites on PKR mRNA abundance in A549 cells. Five of the seven editing sites showed significantly higher normalized mCherry expression compared to their unedited counterparts (Fig. 6e, Additional file 2: Fig. S8B). To assess the collective impact of multiple RNA editing sites on PKR mRNA abundance, we measured endogenous PKR expression in A549 cells upon ADAR1 or ADAR2 KD. We first confirmed that the 3′ UTR editing sites in PKR were edited endogenously in A549 cells. Importantly, these editing sites are mainly regulated by ADAR1 instead of ADAR2 (Additional file 2: Fig. S8C). Upon ADAR1 KD, PKR expression level was significantly reduced by about 40% (Fig. 6f). In contrast, PKR expression did not change upon ADAR2 KD, as expected. These results suggest that the editing sites enhanced PKR mRNA abundance, consistent with the positive editing-expression correlation in primary tumors.

Based on the eCLIP data, the five editing sites that individually promoted PKR mRNA abundance are located within ILF3 binding sites (Fig. 6g, Additional file 2: Fig. S8D-E). To test the hypothesis that ILF3 regulates PKR mRNA abundance in an editing-dependent manner, we generated ILF3 KD A549 cells (Fig. 7a). The edited and unedited reporters, demonstrating differential expression in control cells, no longer produced different expression levels upon ILF3 KD (Fig. 7b). Together, our data suggest that ILF3 promotes PKR mRNA expression in an editing-dependent manner by binding to the PKR mRNA.

### ILF3 knockdown induced EMT in A549 cells

Since ILF3 was found to stabilize transcripts that were differentially edited between E and M tumors, we next asked if ILF3 regulates the EMT process. We carried out ILF3 KD experiments via two different siRNAs in A549 cells. Upon ILF3 KD, cell morphology changed from tightly connected, round cells towards more dispersed, spindle-shaped cells (Fig. 7c), consistent with expected EMT phenotypes. Additionally, we observed reduced expression of the epithelial marker E-cadherin along with increased expression of the mesenchymal marker N-cadherin in the ILF3 KD cells (Fig. 7d, e for protein and RNA levels, respectively). Thus, these data show that ILF3 deficiency induces EMT in A549 cells, supporting a significant role of ILF3 in regulating EMT.

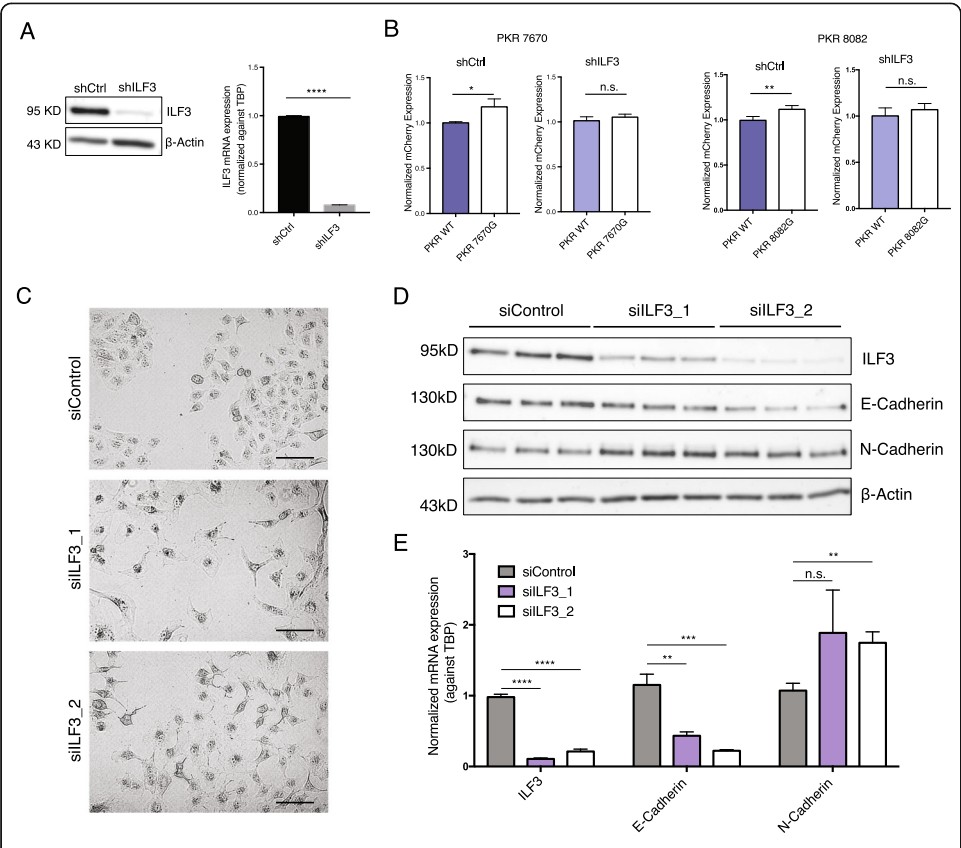

**Fig. 7** ILF3 regulates PKR mRNA abundance and EMT in A549 cells. **a** Western blot confirming shRNA-mediated ILF3 KD in A549 cells (left). ILF3 mRNA levels (mean ± SD) were quantified in A549 shCtrl and ILF3 KD cells by qRT-PCR (right). ILF3 mRNA expression was normalized against gene TBP mRNA expression. Three biological replicates were performed. $p$ value calculated via $t$-test, ****$p < 0.0001$. **b** Normalized mCherry expression (mean ± SD) for nonedited or edited versions of sites in the 3′ UTR of PKR in shCtrl or ILF3 KD A549 cells. Five biological replicates were performed. Normalized expression values were compared between edited and nonedited versions by two-sided $t$-test. *$p < 0.05$, **$p < 0.01$, n.s., not significant. **c** Images of A549 cells transfected with siRNAs targeting ILF3 (two different siRNAs were used to KD ILF3, siILF3_1, and siILF3_2) or control siRNAs (siControl). Scale bars: 100 μm. **d** Western blot detecting protein levels of ILF3, E-Cadherin, N-Cadherin, and internal control β-Actin in the siControl, siILF3_1, and siILF3_2 A549 cells. Three biological replicates were carried out for each experiment. **e** Normalized mRNA expression levels (mean ± SD) for ILF3, E-Cadherin, and N-Cadherin in the siControl, siILF3_1, and siILF3_2 A549 cells. Three biological replicates were carried out for each experiment. The expression values were compared between siILF3 and siControl via $t$-test. **$p < 0.01$, ***$p < 0.001$, ****$p < 0.0001$, n.s., not significant

## Discussion

As most cancer patient deaths are due to metastasis, thorough understanding of the molecular mechanisms underlying metastasis is crucial to developing effective preventative measures [62]. EMT plasticity is thought to underlie cell dissemination and metastatic formation in many cancer types [18]. Supported by studies on primary tumors and various model systems, features of EMT have been associated with metastasis [16, 18, 63, 64]. For instance, higher expression of mesenchymal markers, with preserved epithelial markers in the absence of nearly all canonical EMT transcription factors, was detected in cells located at the leading edge of primary human HNSC tumors [64]. Furthermore, this partial EMT program was correlated with multiple metastatic characteristics, including abundance of lymph node metastases, lymphovascular invasion, and tumor grade [64]. While mutations are understood to drive primary

tumorigenesis and are often found in reported oncogenes and tumor suppressor genes, the existence of recurrently mutated genes specific to metastasis is not clear [18]. Accordingly, mechanisms regulating cell invasiveness beyond genetic variation need to be more thoroughly investigated. Our study is the first to report a systematic characterization of RNA editing in EMT phenotypes across several cancer types. Through a combination of experimental and computational analyses, we observed many editing differences in EMT-relevant genes, especially those related to immune and viral response, with the potential of affecting mRNA abundance of these genes. We also show that higher expression levels of these edited transcripts may be due to stabilization by ILF3.

Located in noncoding regions, most editing sites have unknown function. To assess the contribution of differential editing to altered cell phenotypes in cancer, we focused on the capacity of editing to regulate host gene mRNA abundance. To our knowledge, very few studies have examined this question on the transcriptome-wide scale [65, 66]. Previously, several studies demonstrated this regulatory role for a handful of editing sites through alteration of miRNA binding sequences or mRNA secondary structure or otherwise unknown mechanisms [6, 20, 67–72]. Expanding on these previous studies, we incorporated tissue-rich data from GTEx and ADAR KD expression changes from five cell lines to computationally support associations of editing with mRNA abundance. We also validated the effects of specific editing sites and explored the involvement of RBPs in this regulatory mechanism. It should be noted that we were able to detect associations between editing and mRNA abundance levels, even though differentially expressed genes did not significantly overlap differentially edited genes. These findings do not contradict each other because editing levels are relatively low. Consequently, inosine may affect mRNA abundance, but when present at low levels, may not necessarily lead to significant expression differences.

Considering tumor heterogeneity and the roles of stromal and immune cells in EMT, it is important to examine the contributions of different cell types to differential editing observed in the E-M comparisons. Our results using single-cell data supported that cancer cells are a main cell type underlying differential editing between E and M phenotypes in lung cancer, although contributions by other cell types cannot be excluded. Furthermore, cancer cells demonstrated the strongest expression correlation between ILF3 and immune-relevant differentially edited genes among all cell types considered in lung cancer. These findings suggest that RNA editing is likely an important aspect of transcriptome remodeling of cancer cells in EMT, at least in lung cancer. Single-cell analysis of RNA editing in other cancer types should be conducted in the future.

Our cell line experiments showed EMT induction upon KD of either ADAR1 or ADAR2 in lung and breast cell lines. In contrast, we observed hyperediting in M tumors of most cancer types. The seemingly opposite trends may reflect the complexity of tumor biology that is not effectively recapitulated by cell culture models. Although the cell culture models can support the likely importance of RNA editing in EMT, the exact mechanisms and related regulation can only be investigated using in vivo models in the future. In addition, we did not observe large differences in ADAR expression levels that are consistent with observed editing differences between E and M tumors for all cancer types. Other proteins that directly or indirectly affect ADAR function likely contribute to the regulation of E-M RNA editing differences, which remains to be investigated.

RNA editing is known to be important to innate immunity by preventing viral dsRNA sensors, such as MDA5 and RIG-I, from sensing host dsRNA [35, 39, 73]. In this study, we provided multiple lines of evidence to support that RNA editing differences in EMT may affect immune response genes directly, adding a new dimension to the relationships between RNA editing and innate immunity. Interestingly, a major RBP that mediates this relationship is ILF3. ILF3 was identified as a PKR substrate and serves as a negative regulator of viral replication upon phosphorylation [57, 74]. Upon viral infection and sensing of viral dsRNA, PKR activates, suppresses translation, and promotes apoptosis of affected cells [61]. Importantly, this mechanism has been targeted in oncolytic virotherapy for cancer. Cancer cells that have low PKR expression are sensitive to oncolytic viruses [75–77]. Our study showed that ILF3 mediates the RNA editing-dependent regulation of PKR expression. We also observed that ILF3 KD induced EMT in A549 cells. These data reveal novel insights into the reciprocal regulation between PKR and ILF3 and their potential contributions to EMT. Additional studies on their interaction during viral infection or cancer treatment will also be informative for therapeutic development. Previously, ADAR1 loss has been shown to render tumor cells sensitive to immunotherapy through enhanced inflammatory response [78, 79]. Our findings on the regulation of immune response genes by RNA editing may add additional mechanisms in this process that will need further investigation.

The functional roles of RNA editing in cancer have been increasingly recognized in recent years. Highlighting the extensive editing differences between EMT phenotypes and their impact on mRNA abundance, especially for genes involved in the immune response, our work extends the basis for future studies on the contribution of editing to metastasis and patient outcomes.

## Methods

### Plasmid construction

For bi-directional reporters, full-length or partial 3′ UTR regions (1~2 kb) of candidate genes were cloned from the genomic DNA extracted from HMLE or A549 cells. Edited versions of 3′ UTR inserts were generated using overlap-extension PCR (Supplementary Table 3). Edited and unedited versions of 3′ UTR regions were then cloned into the pTRE-BI-red/yellow vector via ClaI and SalI-HF enzyme sites [48]. To obtain a lentiviral vector expressing ILF3 shRNA, oligos containing the target sequence (GGTCTT CCTAGAGCGTATAAA, TRCN0000329788) were ordered from Integrated DNA Technologies (IDT) and cloned into pLKO.1 via EcoRI and AgeI enzyme sites.

### Cell culture and transfection

A549, Hela, and HEK293T cells were maintained in DMEM with 10% FBS and antibiotic-antimycotic reagent (Gibco). MCF10A cells were maintained in DMEM/F12, supplemented with 5% Horse serum, 20 ng/ml human EGF (PeproTech), 0.5 mg/ml hydrocortisone (Sigma), 100 ng/ml cholera toxin (Sigma), 10 μg/ml insulin (Sigma), and antibiotic-antimycotic reagent (Gibco). For siRNA treatment, A549 or MCF10A cells were seeded at $1 \times 10^5$ cells per well in 6-well plates. After 24 h, siRNAs (Supplementary Table 3) were introduced at the final concentration of 10~100 nM using lipofectamine RNAiMAX (Invitrogen) according to the manufacturer's protocol. Media were

changed 24 h post-transfection, and cells were harvested 72 h post-transfection. For transfection of bi-directional reporters, Hela and HEK293T cells were seeded in 12-well plates to reach 90% confluency by the time of transfection. A549 cells were seeded at $0.15 \times 10^5$ cells per well in 12-well plates 24 h before transfection. Reporter plasmids were transfected at 200 ng per well with lipofectamine 3000 (Invitrogen), following the manufacturer's protocol. Cells were harvested 16 h post-transfection.

### Western blot

Cells were lysed with RIPA buffer containing protease inhibitor (EDTA-free, Thermo Fisher Scientific) at 4 °C for 30 min. The whole cell lysates were then centrifuged at 12, 000*g*, 4 °C for 15 min. The supernatants were collected for protein concentration measurement using Bradford assay (Pierce™ Detergent Compatible Bradford Assay Kit, Thermo Fisher Scientific). Protein samples were prepared by mixing protein lysates with 4× SDS protein loading dye at 3:1 ratio. The mixture was boiled for 5 min. Ten micrograms of each protein samples was loaded on SDS-PAGE gels and transferred to nitrocellulose membranes for antibody incubations. Antibodies used were as follows: ADAR1 antibody (Santa Cruz Biotechnology, sc-73408, 1:200), ADAR2 antibody (Santa Cruz Biotechnology, sc-73409, 1:200), E-cadherin antibody (Cell Signaling Technology, #3195, 1:1000), γ-Catenin antibody (BD Transduction Laboratories, 610253, 1:8000), N-cadherin antibody (BD Transduction Laboratories, 610920, 1:500), Vimentin antibody (Cell Signaling Technology, 5741, 1:1000), NF90(ILF3) antibody (BETHYL Laboratories, A303-651A, 1:1000), β-actin-HRP antibody (Santa Cruz Biotechnology, sc-47778, 1: 2000), goat anti-rabbit IgG-HRP (Santa Cruz Biotechnology, sc-2004, 1:2000), goat anti-mouse IgG-HRP (Santa Cruz Biotechnology, sc-2005, 1:2000). Membrane blots were incubated with SuperSignal West Pico PLUS Chemiluminescent Substrate (Thermo Fisher Scientific) and visualized under the imager (Syngene PXi). Uncropped western blot images are provided in Additional file 2: Fig. S9–13.

### RNA isolation and real-time qPCR

Cells were lysed using TRIzol (Thermo Fisher Scientific). Total RNA was isolated using Direct-zol RNA Miniprep Plus kit (Zymo Research) following the manufacturer's protocol. In total, 2 μg of total RNA was used for cDNA synthesis with SuperScript IV (Thermo Fisher Scientific). The real-time qPCR reaction was assembled using the PowerUp™ SYBR® Green Master Mix (Thermo Fisher Scientific). Primers used for qPCR are listed in Supplementary Table 3. The reaction was performed in the CFX96 Touch Real-Time PCR detection system (Bio-Rad) with the following settings: 50 °C for 10 min, 95 °C for 2 min, 95 °C for 15 s, 60 °C for 30 s, and with the last two steps repeated for 45 cycles. For bi-directional reporter assays, mCherry expression was normalized against eYFP expression within the same sample. ILF3 expression was normalized against the expression of internal control gene *TBP*. For qPCR validating the eCLIP peaks, the final libraries were diluted to the same concentration at 0.01 ng/μl. Five microliters of diluted libraries was used in each qPCR reaction. Around 80 bp upstream each EIF2AK2 editing site was amplified. The expression of each EIF2AK2 region was normalized against the expression of 18s.

### Quantification of RNA editing levels by Sanger sequencing

Regions of interest were amplified from cDNA using Thermo Scientific™ DreamTaq™ Green PCR Master Mix (2X). Primers used for PCR are listed in Supplementary Table 3. The amplicons were gel extracted and premixed with the reverse primer for Sanger sequencing. The peak signals of A and G nucleotides were measured by 4Peaks for editing level calculation (G/(A + G)).

### Categorization of tumors as epithelial and mesenchymal

We downloaded fragments per kilobase million (FPKM) data of primary tumors from patients across seven cancer types in TCGA: BRCA, LUAD, LUSC, PRAD, OV, KIRC, and HNSC, from the Genomic Data Commons (GDC) Data Portal [80]. To assess E and M phenotypes of the tumors of each cancer type, we quantified the enrichment of E and M gene sets by applying gene set variation analysis (GSVA) [81]. We obtained pan-cancer E and M gene sets from a 2014 publication by Tan and colleagues (Table S1A from their publication) [30]. Tumors with high E scores and low M scores were considered to have an E phenotype, while tumors with low E and high M scores were classified as M. Subsets of E and M tumors were selected for each cancer type to minimize confounding of E and M distinction by patient and sample metadata.

### Quantification and comparison of RNA editing levels in TCGA tumors

We downloaded RNA-seq fastq files of categorized tumors from the GDC Legacy Archive. We mapped reads to hg19 with HISAT2, using default parameters. Dense clusters of editing sites, or hyperedited regions, can lead to many mismatches in reads. Consequently, these reads may be left unmapped and hinder accurate detection of editing in these regions. To rescue reads that were originally unmapped due to high density of editing activity, we applied a hyperediting pipeline and combined the recovered reads with uniquely mapped reads for downstream analyses [32, 82]. To analyze editing sites of high confidence, we downloaded the REDIportal database, comprising over 4 million editing sites identified across 55 tissues of 150 healthy humans from GTEx [33, 83]. We applied methods used in our previous studies to detect editing at REDIportal sites in the tumor samples. We filtered out editing sites found in dbSNP (version 147) and COSMIC (version 81), except for reported cancer-related editing sites [8, 13, 19, 84–87], since editing sites have been shown to be mistakenly recorded as SNPs [88, 89]. Within each sample, we also filtered out editing events that overlapped with sample-specific somatic mutations and copy number variants. Somatic variants were obtained from the publicly released MC3 MAF [90], and copy number variants were obtained from copy number segment data downloaded from the GDC data portal.

Differential editing sites were defined as editing sites with significantly different editing levels between E and M phenotypes. To identify such sites, we used an adaptive coverage approach [32]. For an individual editing site, we determined the highest read coverage threshold that was satisfied in at least five samples of both phenotypes, among twenty, fifteen, and ten reads. If none of these thresholds was satisfied and fewer than ten samples in each phenotype had at least five reads covering the site, we did not test the site for differential editing. Using the highest coverage determined, we calculated the mean editing levels among samples of each phenotype separately. We then

consecutively lowered the read coverage threshold by 5 reads and compared the new mean editing levels of each phenotype, when including additional samples, to the original high-coverage-only editing means. If the differences in mean editing levels were less than 0.03, we used the lower read coverage threshold to delineate which samples to include for the differential test. Editing levels between E and M samples were compared by a Wilcoxon rank-sum test. Editing differences were considered significant if the Wilcoxon $p$ value $< 0.05$ and the magnitude of the difference $\geq 0.05$. To account for false positives, we shuffled phenotype labels and retested for significant differences for each differential editing site, 100 times. If a site showed significant differences for shuffled labels over ten times, it was filtered out and no longer considered a differential editing site.

### Identification of differentially expressed genes

HTSeq-Count data were downloaded from the GDC data portal. We identified genes with significantly different mRNA expression levels between E and M tumors of each cancer type, using limma-voom [91]. Metadata significantly correlated with the top two principal components of expression were included as covariates in the linear models. Expression differences were considered significant if log2-fold change was at least 1 and adjusted $p$ value was less than 0.05.

### Rank-rank hypergeometric overlap

To measure the similarity in patterns of editing changes across cancer types, we ranked genes based on differential editing between E and M phenotypes for each cancer type. More specifically, the ranking metric was the statistical significance of the differential editing test (–log10(Wilcoxon $p$ value)), multiplied by the sign of the editing difference (mean of M editing levels – mean of E editing levels). Accordingly, genes at the top of the ranked list had the highest increases in editing in M, while genes at the bottom had the largest decreases in editing in M. For each gene with multiple editing sites tested, the site with the most significant change in editing levels was used to represent the gene. We used the RRHO package within Bioconductor in R to test for significance of overlap between ranked gene lists, with a step size of 30 genes between each rank [92].

We also ran RRHO between gene rankings by differential editing and differential gene expression for each cancer type. To order genes based on differential gene expression, genes were ranked according to the signed statistical significance of differential expression tests (signed by the direction of expression change in M). As a result, genes at the top of the list were more highly expressed in M and genes at the bottom, more lowly expressed in M.

To make RRHO maps comparable across cancer types and across overlaps based on differential editing and differential expression, we scaled the log-transformed $p$ values to account for different lengths of gene lists and then applied the Benjamini-Yekutieli correction for multiple testing [34].

### Gene ontology enrichment analysis

To evaluate whether an individual GO term was enriched in differential editing in one cancer type, we compared the occurrence of the term among query genes—genes

containing differential editing sites—to its occurrences within 10,000 sets of control genes. In each set, one control gene for each query gene was randomly selected among non-differentially edited genes that matched the query gene based on gene length and GC content (within 10%). Query genes that did not have at least ten matched control genes were excluded. We calculated the $p$ value of the term's enrichment among query genes from the normal distribution fit to occurrences of the term among control gene sets. We repeated this assessment of GO term enrichment separately for lists of differential hyperedited and hypoedited genes in each cancer type.

Likewise, we tested the occurrence of each GO term represented among differentially expressed genes to its occurrences among 10,000 sets of non-differentially expressed control genes, randomly selected to match the differentially expressed query genes for gene length and GC content.

### scRNA-seq dataset analysis

We downloaded fastq files from 15 tumor samples of five NSCLC patients [93] and ran CellRanger (version 3.0.2) to map reads and obtain count matrices. We excluded the tumor samples from three LUSC patients exhibiting low percentages of valid barcodes and mapped reads. For the remaining samples, we loaded the filtered feature-barcode matrices from CellRanger and merged the datasets into a single Seurat object with the R package Seurat [94] (version 3.0.2). Next, we filtered out cells that did not meet the following criteria: 101–6000 expressed genes, over 200 UMIs, and less than 10% UMIs corresponding to the mitochondrial genome. Following normalization by sctransform [95] (version 0.2.0), we performed dimensional reduction with PCA. Based on an elbow plot, we decided to consider the first ten PCs for downstream clustering and TSNE embedding. To assign cell identity labels to clusters, we matched differentially expressed genes of clusters to reported marker genes. One cluster had differentially expressed markers of multiple cell types, so we subclustered its cells. To assess the accuracy of our final labeling of nine cell types, we examined expression of marker genes across the cell types in two approaches. In one approach, we used CIBERSORTx [96] to generate a gene expression signature matrix, which is a matrix of expression signatures characterizing cell types. To create this matrix from expression profiles of single cells labeled by cell type, CIBERSORTx identified differentially expressed genes. In the second approach, we pooled reads of each cell type together and calculated RPKM. These RPKM values calculated from pooled cells were also used to correlate ILF3 expression with expression of editing-correlated genes.

To identify cancer cells with E and M phenotypes, we subclustered the cancer cells. To this end, we first ran sctransform and PCA on only the cancer cells. Using the first twelve PCs, we clustered the cells and performed non-linear dimension reduction by UMAP. As a cluster of 200 M cells was identified, we sampled 200 E cells with similar numbers of features, numbers of UMIs, and percentages of reads mapped to the mitochondrial genome. For each phenotype, we compiled reads of cells together and detected editing levels at REDIportal sites. For each testable editing site, E and M editing levels were compared by Fisher's exact test. An editing site was considered differential if the difference in editing levels was at least 0.05 and Fisher's exact $p$ value $< 0.05$.

### RNA-seq generation for ADAR KD A549 cells

A549 cells were seeded at $1 \times 10^5$ cells per well in 6-well plates 24 h before siRNA transfection. siRNAs (Supplementary Table 3) were introduced at the final concentration of 22 nM using lipofectamine RNAiMAX (Invitrogen), according to the manufacturer's protocol. For individual KD of ADAR1 or ADAR2, 11 nM siRNA of ADAR1 or ADAR2 were mixed with 11 nM control siRNAs. For double KD of ADAR1 and ADAR2, 11 nM siRNA of ADAR1 and 11 nM siRNA of ADAR2 were mixed. Media were changed 24 h post-transfection. The transfected cells were harvested 48 h post-transfection. Total RNA was extracted for RNA-seq library generation for three biological replicates of each condition. RNA sequencing libraries were generated using NEBNext Ultra II Directional RNA library Prep kit and NEBNext multiplex oligos for Illumina according to the manufacturer's instructions (New England Biolabs, E7760S). Library concentrations were measured by Qubit fluorometric assay (Life Technologies), and libraries were sequenced on an Illumina HiSeq-4000 with 150-bp paired-end reads.

### A549 ADAR KD RNA-seq analysis

Following mapping of RNA-seq reads with HISAT2 and a hyperediting pipeline [32], we detected editing events at REDIportal sites as we did for the TCGA tumor samples. We then removed dbSNP variants while retaining previously reported cancer editing sites. To identify differential editing sites between each ADAR KD condition and control or between each individual ADAR KD and double KD, we used REDIT-LLR on sites that were edited in the control condition (editing level $\geq 0.05$) [97]. A site was considered differentially edited if the difference in mean editing levels between conditions was at least 0.05 and REDIT-LLR $p$ value $< 0.05$.

### Regression analysis

For each differential editing site, association between editing level and host gene mRNA abundance was tested by fitting a linear model of log-transformed gene FPKM against editing level and potentially confounding covariates (using the lm function in R). For associations in GTEx data, we included age, gender, and race as covariates. For associations in TCGA data, we included metadata that were significantly correlated with the top two principal components of expression, as in the differential expression analysis.

### eCLIP-seq generation

Following a published protocol [53], we performed an eCLIP experiment comprising three libraries from two ILF3-immunoprecipitated biological replicates and one control. The antibody used for this experiment is ILF3/NF90 antibody (Bethyl Laboratories, A303-651A). For each sample, 10M A549 cells were ultraviolet (UV) crosslinked at 254 nm (800 mJ cm$^{-2}$). We then performed cell lysis, RNA fragmentation, immunoprecipitation, adapter ligation, and other library preparation steps on UV crosslinked samples, as described [53]. For the size-matched input control (SMInput), we prepared a library from sampling 2% of one pre-immunoprecipitation UV crosslinked sample. This control is used to normalize binding signal, given biases that may be introduced through various experimental steps.

## eCLIP-seq peak calling and distance analysis

We obtained eCLIP peak data for 96 RBPs in K562, 83 RBPs in HepG2, and ILF3 in A549 cells, as described previously [46]. Briefly, after demultiplexing and trimming adapters, we aligned reads in multiple rounds with STAR. First, reads aligning to rRNA sequences were discarded, and then the unmapped reads were aligned to Alu sequences, permitting a maximum of 100 alignments for an individual read. In the final alignment step, the remaining unmapped reads were uniquely aligned to the hg19 genome. Then read enrichment within a sliding window, considering both genome and Alu-aligned reads, was tested for significance by a Poisson model in order to call eCLIP peaks [46, 98].

To assess the proximity of a single RBP's binding to differential editing sites compared to random controls, we calculated the distance from each differential editing site or control to the closest eCLIP peak in the same gene. Control sites consisted of adenosines within genes containing differential editing sites [32]. We then calculated the area under the curve (AUC) of the cumulative distribution of distances from differential editing sites to the closest eCLIP peaks. Given our interest in close binding, we considered distances up to 10,000 bases only for AUC calculation. Similarly, we calculated the AUC of the distribution of closest distances between eCLIP peaks and controls, for each of 10,000 sets of random controls. We computed the $p$ value of the AUC for differential editing sites from the normal distribution fit to the AUC values of control sets [32].

## Supplementary information

---

**Additional file 1:** Table S1. Primary tumor samples used in this study. Cancer types and the corresponding numbers of categorized E and M tumor samples analyzed in this study. Table S2. List of editing sites predicted to regulate host gene mRNA abundance. Editing-expression associations were supported by consistent expression changes upon ADAR KD in at least one cell line. editlevel_est represents editing level regression coefficient; adj_edit_pvalue is the adjusted $p$-value of the coefficient. Table S3. List of primers and siRNAs used in this study.

**Additional file 2:** Fig. S1. Differential editing not confounded by metadata. Heatmaps of significance (log10-transformed adjusted $p$ values) of correlations between the top two principal components and E/M phenotype among metadata fields in each cancer type. Darker color indicates smaller $p$ value and stronger association. Fig. S2. Gene ontology enrichment among differentially edited genes. Significance of enrichment of gene ontology (GO) terms among all differentially edited genes (blue), only hyperedited genes (green) or only hypoedited genes (pink) of each cancer type. Point size represents the statistical significance of enrichment (log10-transformed adjusted $p$ value). Terms significantly enriched in at least two cancer types are shown. For cancer types with a global hyperediting trend in M tumors, GO enrichment among hyperedited genes is similar to that among all differentially edited genes. Likewise, for cancer types with a hypoediting trend (BRCA and OV), enrichment among hypoedited genes is similar to that among all differentially edited genes. Fig. S3. Clustering of single cells from three lung cancer tumors. A. TSNE projection of cells based on expression profiles, with color indicating cluster identity (left). Cell types were assigned to clusters by matching differentially expressed genes of clusters to known cell type markers (right). B. TSNE projection of only cells from cluster 10 to further refine cell type assignment (left). Similar to A, cell types were labeled using differentially expressed genes that matched cell type markers (right). C. Counts of cells for each cell type after 2 rounds of clustering and cell type assignment (A and B). D. Log2-transformed expression values of marker genes across cell types. Signature matrix on the left indicates expression values assigned for each cell type by CIBERSORTx. On the right, Pooled Cells indicate that expression values were calculated from pooling reads from cells of the same type together. Fig. S4. E and M assignment of single cells not confounded by metadata. Comparison between E and M cells altogether (top) and within each tumor sample (bottom) of metadata fields: UMI count (A-B), gene count (C-D), and percent of reads mapping to the mitochondrial genome (E-F). Metadata values were compared by Mann Whitney U tests, and significance of $p$ values are shown. ns: $p > 0.05$, * $p \leq 0.05$, ** $p \leq 0.01$. Fig. S5. LUAD and LUSC tumor editing differences of differential sites identified from single cell RNA-seq analysis. For each editing site, the difference in mean editing levels between M and E tumors (M - E) in each cancer type is listed. Green highlight indicates Wilcoxon $p$ value < 0.05. Fig. S6. Altered editing upon knockdown of ADAR1, ADAR2, or both. A. Distributions of mRNA expression of ADAR1 and ADAR2 under ADAR KD and control conditions. Expression levels were quantified as transcripts per million (TPM). B. Mean editing levels of testable sites in five comparisons between ADAR KD conditions or control experiment. Sites with significant editing differences between conditions are colored red, while gray represents nondifferential sites. $Y = x$ line shown in blue. C. Proportions of lung cancer E-M differential sites that were also differential in ADAR KD conditions (compared to controls). sigADAR1: sites that were differential only in ADAR1 KD. sigADAR2: sites that were differential only in ADAR2 KD. sigBoth: sites that were differential in both ADAR1 KD and ADAR2 KD, or in double KD. The

prefix 'red' indicates reduced editing level by at least 0.05 upon KD from control, but did not pass the statistical significance requirement. 'Remain': editing sites that were not significantly different or reduced across any comparison. Fig. S7. Expression of ADARs in E and M tumors. Distributions of mRNA expression of ADAR1 (left) and ADAR2 (right) in E and M tumors across cancer types. Expression values, measured as Fragments Per Kilobase per Million mapped reads (FPKM), were compared by Mann Whitney U tests, and significance of $p$ values are shown. ** $p \leq$ 0.01; *** $p \leq$ 0.001; **** $p \leq$ 0.0001. Fig. S8. ILF3 binds closely to the differential editing sites in editing-expression correlated genes. A. Histogram of distances between differential editing sites in editing-correlated genes and the closest ILF3 eCLIP peaks in HepG2 and K562 cells (turquoise), up to 10 kb. Gray curves represent distances for 10 sets of randomly picked A's in the same genes as differential editing sites. Number of differential editing sites is given by $n$ for each cell line. $p$ value was calculated by comparing the area under the curve (AUC) of the distance distribution for differential editing sites to a normal distribution fit to the AUC values of 10,000 sets of random gene-matched A's. B. Normalized mCherry expression for nonedited or edited versions of sites in the 3' UTR of PKR in A549 cells. Five biological replicates were performed. Normalized expression values were compared between edited and nonedited versions by two-sided t-test. ** $p < 0.01$. C. Editing levels of PKR 3' UTR editing sites in siControl, siADAR1 and siADAR2 A549 cells measured by Sanger sequencing. The peak signals of A and G nucleotides were measured by 4Peaks for editing level calculation (G/(A + G)). The editing level of each editing site (underlined) is shown in the graph. D. Read coverage of ILF3 eCLIP-seq in HepG2 and K562 cells for two biological replicates (ILF3 IP1 and ILF3 IP2, turquoise) and size-matched input (SMInput, gray) in each cell line. The five validated 3' UTR editing sites affecting PKR mRNA abundance in A549 cells are labeled in magenta. E. Validation of PKR eCLIP signal overlapping three editing sites. PKR expression was measured by qRT-PCR in the IP or SMInput samples and normalized against the expression of 18s rRNA. Three technical replicates were performed (other than two replicates for 8034). $p$ value calculated by t-test. * $p < 0.05$, ** $p < 0.01$, **** $p < 0.0001$. Fig. S9. Uncropped western blot images for Fig. 4a. Fig. S10. Uncropped western blot images for Fig. 4b. Fig. S11. Uncropped western blot images for Fig. 4c. Fig. S12. Uncropped western blot images for Fig. 7a. Fig. S13. Uncropped western blot images for Fig. 7d.

**Additional file 3.** Review history.

## Acknowledgements
We thank members of the Xiao and Cheng laboratories for helpful discussions and comments on this work. The results published here are in part based upon data generated by The Cancer Genome Atlas managed by the NCI and NHGRI. We thank the GTEx consortium for generating the RNA-seq data. We thank the ENCODE Project Consortium (specifically the groups of Dr. Gene Yeo and Dr. Brenton Graveley) for generating the eCLIP-seq and RNA-seq data sets used in this study. We appreciate the helpful discussions with Dr. Eric Van Nostrand and Dr. Gene Yeo on the ILF3 eCLIP-seq experiments.

## Review history
The review history is available as Additional file 3.

## Peer review information

## Authors' contributions
T.W.C., T.F., C.C., and X.X. designed the study with inputs from all other authors. T.W.C., J.H.L., and G.Q.V. conducted the bioinformatics analyses. T.F., J.H.B., and H.I.J. conducted the molecular and cellular experiments. All authors contributed to the writing of the paper. The authors read and approved the final manuscript.

## Authors' information
Twitter handles: @gracexiao99 (Xinshu Xiao).

## Funding
This work was supported in part by grants from the National Institutes of Health (U01HG009417, R01AG056476 to X.X. and R35GM131876 to C.C.) and the Jonsson Comprehensive Cancer Center at UCLA. C.C. is a CPRIT Scholar in Cancer Research (RR160009). T.W.C. was supported by the NIH-NCI National Cancer Institute T32LM012424.

## Availability of data and materials
eCLIP-seq and RNA-seq data sets are available at the Gene Expression Omnibus (GEO): https://www.ncbi.nlm.nih.gov/geo/query/acc.cgi?acc=GSE147487 [99]. Other data analyzed in this study are from the GDC data portal at https://portal.gdc.cancer.gov/ [100], the GTEx portal at https://gtexportal.org/home/ [101], the ENCODE project at http://www.encodeproject.org [102], the REDIportal database at http://srv00.recas.ba.infn.it/atlas/ [33], ArrayExpress at https://www.ebi.ac.uk/arrayexpress/experiments/E-MTAB-6149/ [40, 93], and GEO under accession numbers GSE28040 [1] and GSE38233 [47].

## Ethics approval and consent to participate
Not applicable.

## Competing interests
The authors declare no competing interests.

## Author details
[1]Bioinformatics Interdepartmental Program, UCLA, Los Angeles, CA, USA. [2]Molecular, Cellular and Integrative Physiology Interdepartmental Program, UCLA, Los Angeles, CA, USA. [3]Department of Integrative Biology and Physiology, UCLA, Los Angeles, CA, USA. [4]Department of Life and Nanopharmaceutical Sciences & Oral Microbiology,

School of Dentistry, Kyung Hee University, Seoul, South Korea. [5]Department of Bioengineering, UCLA, Los Angeles, CA, USA. [6]Lester & Sue Smith Breast Center & Department of Molecular and Human Genetics, Baylor College of Medicine, Houston, TX, USA. [7]Molecular Biology Institute, UCLA, Los Angeles, CA, USA. [8]Institute for Quantitative and Computational Sciences, UCLA, Los Angeles, CA, USA. [9]Jonsson Comprehensive Cancer Center, UCLA, Los Angeles, CA, USA.

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

## 