## [**Additional file 3.** Review history. · Genome Biology]

Review History

First round of review

Reviewer 1

Are you able to assess all statistics in the manuscript, including the appropriateness of statistical tests used? Yes, and I have assessed the statistics in my report.

Comments to author:

This manuscript provides the first transcriptome-wide analysis of RNA editing differences between epithelial (E) and mesenchymal (M) tumors. The authors should be commended for the very thorough computational analysis performed to both identify editing sites (using multiple mapping approaches as well as multiple SNP databases) and to quantify editing level differences (adaptive coverage approach, as well as phenotype shuffling for E/M classification). The identification of immune response genes as a class of differentially edited genes is an exciting result as a role for RNA editing in preventing aberrant recognition of self dsRNA is an area of intense investigation, but the role of editing events in immune genes is understudied. The roles of ILF3 as a direct regulator (promotion) of RNA editing in the immune response gene, PKR, provides an important mechanistic test of the biology identified from the genomic analyses.

Points that would support the conclusions in the manuscript are listed below:

1. The authors report a global hypoediting trend for M/E BRCA and OV tumors and a hyperediting trend in the LUAD, LUSC, PRAD, KIRC and HNSC M/E tumors. The editing differences in these tumors primarily reside in introns and UTRs, which are normally edited by ADAR1. The authors should report ADAR1 levels between E/M tumors of these cancer types.
2. The scRNA-seq analysis seems to support the mesenchymal hyperediting observed in the bulk analysis of other types of lung cancer. However, it is unclear if the 9 sites identified from the scRNA-seq were also observed as hyperedited in the bulk tumors? The authors should provide a table with the 9 sites/associated genes identified from the scRNA-seq and the editing levels observed in the bulk tumor analysis.
3. The analysis of ADAR1/2 KO on cell phenotypes does not seem to be consistent with the bulk tumor analysis. A549 is a lung epithelial line and MCF10A is a breast epithelial line. The bulk tumor analysis suggest hyper-editing and hypoediting for when comparing M/E for lung and breast cancers, respectively. It is then unclear why loss of ADARs promotes EMT for both lung and breast cell lines. Wouldn't the expectation be that overexpression of ADARs promotes EMT in lung, while ADAR KO promotes EMT in breast? While tumor heterogeneity could be used as a rationale here (as in the Discussion), the scRNA-seq was performed on lung cancer cells and indicated that lung cancer cells exhibit hyperediting for M/E.
4. The results regarding ILF3 binding promoting RNA editing of the PKR 3' UTR and in turn PKR mRNA expression are intriguing. However, some of the effects, such as editing of individual 3' UTR editing events on PKR expression are quite minor (Fig. 6E). One piece of data that would enhance this section is qPCR analysis of PKR in the Control and ILF3 shRNA lines (as was done for ILF3 in Figure 6G).

Minor points with text:

1. Page 5, Line 15. The reference listed is incorrect. The correct reference is Chen et al., RNA, 2000.
2. The cell lines used for the editing 3' UTR reporter assays (Fig 5D) are not mentioned in the text or methods section.
3. The conclusion of the ILF3 section should be more explicit about the conclusions of mRNA and editing regulation. Could the authors revise to state that ILF3 promotes PKR mRNA expression by binding to the PKR mRNA and enhancing RNA editing within the 3'UTR?

Reviewer 2

Are you able to assess all statistics in the manuscript, including the appropriateness of statistical tests used? Yes, and I have assessed the statistics in my report.

Comments to author:

This manuscript by Chan et al reports differential editing sites between epithelial and mesenchymal tumors in seven cancer types using TCGA data. Single-cell RNA sequencing analysis suggests that differential editing happens in the cancer cells other than other cell types in the tumors. The loss of either ADAR protein promotes EMT. The differential editing sites are enriched on immune-related mRNAs. Further, the authors show that ILF3 binding is enriched near these editing sites and enhances the stability of the host mRNAs. Overall, the observation of differential editing between epithelial and mesenchymal tumors and the potential roles of ADARs in EMT is intriguing. However, the mechanism that editing affects mRNA stability through ILF3 is not convincing.

Major concerns:

I suggest that the authors analyze the expression of ADAR proteins in E and M tumors.

The trend of editing difference in 7 types of cancers is the opposite (Fig. 1B). However, the authors found a significant overlap in genes with editing changes among multiple cancer types (Fig. 2A). Do these results suggest that the same group of mRNAs have opposite editing trends in 7 cancer types? If so, how could these opposite editing differences explain the EMT?

There are differential editing sites between E and M tumors, and these differential editing sites affect the host mRNAs abundance. However, according to the data (Fig. 2B, 5D, and 6E), such effect is subtle. There is lack of evidence (experiments or publications) to show subtle alteration of these mRNAs results in EMT.

Fig. 4. Since both ADAR1 and ADAR2 are expressed in A549 and MCF10A cells, to rule out the possibility of compensation between two ADAR proteins, it is necessary to detect the editing level of some sites (differentially edited between E and M) after ADAR knockdown. Besides, the authors may need to perform rescue assays of ADAR loss experiments to solidify their finding.

Fig. 1B shows that editing level increases in M tumor compared to E in both LUAD and LUSC, suggesting that editing has a "pro-EMT" effect. However, Fig. 4 shows that the loss of ADAR promotes EMT in A549 cells, suggesting editing has an "anti-EMT" effect. The conflicting results need to be explained.

Since ILF3 is required to regulate the abundance of mRNAs containing differentially edited sites, it would be very helpful to evaluate if ILF3 ablation affects EMT.

Additional experiments are required to check whether editing will affect ILF3 binding to its target mRNAs.

Minor concerns:

Fig. 2C. The authors may need to perform a GO analysis of all the edited mRNAs as a control. Besides, it would be better to perform the GO analysis on hyper- and hypo- edited gene lists separately.

Fig. 3A. The abbreviation "EC" is not explained - it may be endothelial cells.

Page 6, Line 19-23. The difference in gene expression between different cell types is subtle (Fig. 3A). Thus, this result is not sufficient to conclude that "the editing differences observed among bulk tumors may be predominantly attributable to the cancer cells". The authors need to compare the editing in different cell types in the tumor.

Fig. 3C. I only saw 8 sig points in figure (9 described in the context). The difference seems subtle to me.

Fig. 6A. It would be better to use unchanged editing sites in the same mRNAs as control rather than randomly picked adenosine.

Authors Response

We'd like to thank the reviewers for their constructive and insightful comments. We appreciate the reviewers' recognition of the contributions and significance of this study. In line with the reviewers' comments, we have conducted additional experimental and bioinformatic analyses that substantially strengthen the conclusions presented in the initially submitted manuscript.

Modified sections and sentences related to the response are highlighted in green in the main text.

Reviewer #1:

This manuscript provides the first transcriptome-wide analysis of RNA editing differences between epithelial (E) and mesenchymal (M) tumors. The authors should be commended for the very thorough computational analysis performed to both identify editing sites (using multiple mapping approaches as well as multiple SNP databases) and to quantify editing level differences (adaptive coverage approach, as well as phenotype shuffling for E/M classification). The identification of immune response genes as a class of differentially edited genes is an exciting result as a role for RNA editing in preventing aberrant recognition of self dsRNA is an area of intense investigation, but the role of editing events in immune genes is understudied. The roles of ILF3 as a direct regulator (promotion) of RNA editing in the immune response gene, PKR, provides an important mechanistic test of the biology identified from the genomic analyses.

Points that would support the conclusions in the manuscript are listed below:

1. The authors report a global hypoediting trend for M/E BRCA and OV tumors and a hyperediting trend in the LUAD, LUSC, PRAD, KIRC and HNSC M/E tumors. The editing differences in these tumors primarily reside in introns and UTRs, which are normally edited by ADAR1. The authors should report ADAR1 levels between E/M tumors of these cancer types.

Thank you for this point. ADAR1 mRNA expression levels were slightly higher in M tumors of certain cancer types with a hyperediting trend (Figure R1). We also observed elevated ADAR2 levels in M tumors of most cancer types. While higher expression of these ADARs likely contributed to increased editing in the M phenotype of several cancer types, significant expression differences were absent for some cancer types or in the case of BRCA, inconsistent with the editing trend. Thus, ADAR expression levels do not adequately explain the general hypoediting and hyperediting trends observed in this study. The regulatory roles of other proteins in editing differences between E and M phenotypes should be further investigated in a future study.

We have included the ADAR expression levels as Supplementary Fig 7 and revised the manuscript accordingly (page 7).

Figure R1. Expression of ADARs in E and M tumors. Distributions of mRNA expression of ADAR1 (left) and ADAR2 (right) in E and M tumors across cancer types. Expression values, measured as Fragments Per Kilobase per Million mapped reads (FPKM), were compared by Mann Whitney U tests, and significance of p-values are shown (** $p \leq 0.01$; *** $p \leq 0.001$; **** $p \leq 0.0001$).

2. The scRNA-seq analysis seems to support the mesenchymal hyperediting observed in the bulk analysis of other types of lung cancer. However, it is unclear if the 9 sites identified from the scRNA-seq were also observed as hyperedited in the bulk tumors? The authors should provide a table with the 9 sites/associated genes identified from the scRNA-seq and the editing levels observed in the bulk tumor analysis.

In LUAD and LUSC respectively, only one of the nine sites passed the p-value cutoff ($p < 0.05$, Wilcoxon rank-sum test). However, as shown in the table below, their magnitude of editing change did not exceed 5% to meet our requirement of a significant change. Nonetheless, both sites showed higher editing in M tumors. Thus, the hyperediting trend in M samples is supported by these two sites. We acknowledge that the overlap here is very small and no conclusion should be made based on such a small sample size. This small overlap likely reflects the low coverage on editing sites in the single cell data, and/or the possibility that more differential editing sites exist in the bulk tumors that were not identified in our study due to limits in power.

We have included the above data as Supplementary Fig. 5 in the manuscript (page 6).

	LUAD	LUSC
RP11-792A8.4 chr7:66205084	-0.007	0.012
RHOA chr3:49397323	-0.011	0.043
MRPS16 chr10:75008841	-0.014	-0.019
MRPS16 chr10:75008817	-0.02	-0.029
MRPS16 chr10:75008815	-0.0089	-0.019
MRPS16 chr10:75008797	-0.0082	-0.0011
BPNT1 chr1:220231254	0.02	-0.0082
ARL16 chr17:79648370	0.0032	-0.00036
AC007246.3 chr2:39701980	0.021	0.01

■ $p < 0.05$

Table R1. LUAD and LUSC tumor editing differences of differential sites identified from single cell RNA-seq analysis. For each editing site, the difference in mean editing levels between M and E tumors (M - E) in each cancer type is listed. Green highlight indicates Wilcoxon p-value < 0.05.

3. The analysis of ADAR1/2 KO on cell phenotypes does not seem to be consistent with the bulk tumor analysis. A549 is a lung epithelial line and MCF10A is a breast epithelial line. The bulk tumor analysis suggest hyper-editing and hypoediting for when comparing M/E for lung and breast cancers, respectively. It is then unclear why loss of ADARs promotes EMT for both lung and breast cell lines. Wouldn't the expectation be that overexpression of ADARs promotes EMT in lung, while ADAR KO promotes EMT in breast? While tumor heterogeneity could be used as a rationale here (as in the Discussion), the scRNA-seq was performed on lung cancer cells and indicated that lung cancer cells exhibit hyperediting for M/E.

This is a great point. These observations reflect the complex nature of the potential mechanisms of RNA editing regulation and function in cancer. The fact that ADAR KD in cell culture caused EMT changes suggests that RNA editing changes may drive EMT or at least constitute an important aspect of EMT mechanisms. However, in the primary tumors, we did not observe a high level of difference of ADAR expression levels between E and M tumors that are consistent with the observed RNA editing differences for all cancer types (response to point #1 above). Thus, ADAR expression may not fully account for editome difference between E and M tumors. This difference may be due to regulation by other proteins that directly or indirectly affect ADAR function. We wish to investigate this question in our future work. The seemingly opposite trend in E vs. M RNA editing differences between bulk tumors and cultured cell lines may reflect the complexity of cancer biology that is not effectively recapitulated by cell culture models. Although the cell culture models can support the likely importance of RNA editing in EMT, the exact mechanisms and related regulation can only be investigated using in vivo models in the future. We have now updated the Discussion (page 11) to clarify the above points.

4. The results regarding ILF3 binding promoting RNA editing of the PKR 3' UTR and in turn PKR mRNA expression are intriguing. However, some of the effects, such as editing of individual 3' UTR editing events on PKR expression are quite minor (Fig. 6E). One piece of data that would enhance this section is qPCR analysis of PKR in the Control and ILF3 shRNA lines (as was done for ILF3 in Figure 6G).

Thank you for this point. We carried out the suggested experiment where either an siRNA or an shRNA was used to knockdown (KD) ILF3 in A549 cells, then tested endogenous PKR expression via qRT-PCR (Figure R2A, B below). These experiments showed that the mRNA expression level of PKR was elevated upon ILF3 KD. This result is not consistent with the expectation that ILF3 stabilizes PKR mRNA via the RNA editing sites. However, this observation is not surprising. Indeed, ILF3 KD caused significant upregulation of interferon stimulated genes (ISGs), including ADAR1 p150 (Figure R2C). Thus, it is possible that other mechanisms (e.g., interferon related) caused enhanced PKR expression upon ILF3 KD. Given the diverse

roles of ILF3 [1], there may exist multiple types of regulatory relationships between ILF3 and PKR, with the editing-mediated PKR expression regulation by ILF3 being one of them. Indeed, it is known that, upon viral infection, PKR activates and phosphorylates ILF3, with the latter functioning as a translational repressor of viral RNAs [2]. Future studies are needed to dissect the complete reciprocal relationships between PKR and ILF3.

It should be noted that we observed multiple RNA editing sites in the 3' UTR of PKR. Although the effect of individual sites on PKR expression was small, the combined impact of multiple sites on PKR may be relatively large. To further confirm the impact of RNA editing on PKR, we measured endogenous PKR expression in ADAR1 and ADAR2 KD A549 cells. We first confirmed that the 3' UTR editing sites in PKR were edited endogenously in A549 cells. Importantly, these editing sites are mainly regulated by ADAR1 instead of ADAR2 (Figure R2D). Upon ADAR1 KD, PKR expression level was significantly reduced (~40% reduction) (Figure R2E), consistent with the expectation that RNA editing stabilizes PKR mRNA. In contrast, PKR expression did not change upon ADAR2 KD, as expected (Figure R2E).

We have revised the manuscript to include the above results (Figure R2D-E as new Supplementary Fig. 8C and Figure 6F, page 10). In addition, we have revised the Discussion section to discuss the complex relationships between ILF3 and PKR (page 12).

Figure R2. PKR expression upon ILF3 or ADAR KD in A549 cells. **A.** Normalized mRNA expression levels for ILF3 and PKR in shRNA-mediated ILF3 (shILF3) KD A549 cells and control A549 cells (shCtrl). ILF3 mRNA expression was measured by qRT-PCR and normalized against gene TBP mRNA expression. Error bars: standard deviation (same below). Three biological replicates were carried out for each sample and the gene expression values were compared between shCtrl and shILF3 A549 cells by two-sided t-test. * $p < 0.05$, **** $p < 0.0001$. **B.** Normalized mRNA expression levels for ILF3 and PKR in siRNA-mediated ILF3 KD A549 cells (two different siRNAs were used for ILF3 KD: siILF3 1, siILF3 2) and control A549 cells (siControl). Three biological replicates were conducted for each sample and the gene expression values were compared between siILF3 and siControl A549 cells by two-sided t-test. ** $p < 0.01$, **** $p < 0.0001$. **C.** Normalized mRNA expression levels of ADAR1 p150 isoform, OAS1, IFIT1, and MDA5 in the siControl, siILF3_1 and siILF3_2 A549 cells. Three biological replicates were conducted for each sample. Gene expression values were compared between siILF3 and siControl A549 cells by two-sided t-test. * $p < 0.05$, ** $p < 0.01$, *** $p < 0.001$, **** $p < 0.0001$. **D.** Editing levels of PKR 3'UTR editing sites in the siControl, siADAR1 and siADAR2 A549 cells measured by Sanger sequencing. The peak signals of A and G nucleotides were quantified by 4Peaks for editing level calculation ($G/(A+G)$). The editing level of each editing site (underlines) is shown in the graph. **E.** Normalized mRNA expression levels for PKR in the siControl, siADAR1 and siADAR2 A549 cells. Three biological replicates were conducted for each sample. Gene expression values were compared between siADAR1/2 and siControl A549 cells by two-sided t-test. * $p < 0.05$, n.s, not significant.

References:

1. Castella, Sandrine, et al. "Ilf3 and NF90 functions in RNA biology." Wiley Interdisciplinary Reviews: RNA 6.2 (2015): 243-256.
2. Harashima, A., Guettouche, T. and Barber, G.N. (2010) Phosphorylation of the NFAR proteins by the dsRNA-dependent protein kinase PKR constitutes a novel mechanism of translational regulation and cellular defense. *Genes Dev.*, 24, 2640–2653.

Minor points with text:

1. Page 5, Line 15. The reference listed is incorrect. The correct reference is Chen et al., RNA, 2000.

Thank you, we have corrected this reference.

2. The cell lines used for the editing 3' UTR reporter assays (Fig 5D) are not mentioned in the text or methods section.

Thank you for this point. All 3'UTR reporter assays in Fig. 5D were conducted in HeLa cells. We have updated the legend of Fig. 5D to include this information.

3. The conclusion of the ILF3 section should be more explicit about the conclusions of mRNA and editing regulation. Could the authors revise to state that ILF3 promotes PKR mRNA expression by binding to the PKR mRNA and enhancing RNA editing within the 3'UTR?

We have added a similar concluding sentence at the end of the relevant ILF3 section (page 10).

Reviewer #2:

This manuscript by Chan et al reports differential editing sites between epithelial and mesenchymal tumors in seven cancer types using TCGA data. Single-cell RNA sequencing analysis suggests that differential editing happens in the cancer cells other than other cell types in the tumors. The loss of either ADAR protein promotes EMT. The differential editing sites are enriched on immune-related mRNAs. Further, the authors show that ILF3 binding is enriched near these editing sites and enhances the stability of the host mRNAs. Overall, the observation of differential editing between epithelial and mesenchymal tumors and the potential roles of ADARs in EMT is intriguing. However, the mechanism that editing affects mRNA stability through ILF3 is not convincing.

Major concerns:

I suggest that the authors analyze the expression of ADAR proteins in E and M tumors.

We appreciate this suggestion. Measured by reverse phase protein assay (RPPA), protein expression of ADAR1 was available in the TCGA data for a small number of E and M tumors in a few cancer types (Figure R3A below). We analyzed the protein level difference in three cancer types with at least 20 samples in the E or M group: LUAD, LUSC and PRAD. No significant difference in normalized ADAR1 protein expression was observed between E and M tumors in LUSC and PRAD, while a slightly lower expression in M tumors was observed in LUAD (Figure R3B). We are concerned that the small sample size here with protein level quantification may not provide robust results. ADAR2 protein levels are not available at TCGA.

Next, we examined mRNA expression of ADARs using the RNA-seq data (as also suggested by reviewer 1, point 1). The result is included in this document as Figure R1 (response to reviewer 1) and as Supplementary Fig. 7 in the revised manuscript. Although a statistically significant difference of ADAR expression levels was observed for some cancer types that are consistent with the E-M editing differences, such significance was not observed for all cancer types. Although statistically significant, the magnitudes of ADAR expression changes were relatively small. Thus, ADAR expression may not adequately explain the observed RNA editing differences. The RNA editing difference between E and M may be due to regulation by other proteins that directly or indirectly affect ADAR function. We wish to investigate this question in our future work.

We have now revised the manuscript to include the above data and related discussions (pages 7, 11)

A

Cancer Type	E Count	M Count
BRCA	0	0
LUAD	24	27
LUSC	33	22
PRAD	65	74
OV	1	1
KIRC	2	2
HNSC	0	0

B

Figure R3. A. Available ADAR1 protein expression data. For each cancer type analyzed, the numbers of E and M tumors with ADAR1 protein expression levels from RPPA, are listed. **B.** Protein expression of ADAR1 measured by RPPA. Distributions of normalized ADAR1 protein levels in E and M tumors across cancer types with available data. Normalized expression values were compared by Mann Whitney U tests, with significance of p-value ** indicating $p \leq 0.01$.

The trend of editing difference in 7 types of cancers is the opposite (Fig. 1B). However, the authors found a significant overlap in genes with editing changes among multiple cancer types (Fig. 2A). Do these results suggest that the same group of mRNAs have opposite editing trends in 7 cancer types? If so, how could these opposite editing differences explain the EMT?

Thank you for bringing up this question. The RRHO map shows that the highest overlaps are among genes with most significant editing changes in the same direction. That is, the regions with strongest significance were found at the bottom left (hyperedited in both cancer types) or top right (hypoedited in both) corners within pairs of cancer types. We did not observe such significance at the top left or bottom right corners (hyperedited in one and hypoedited in the other), suggesting weaker overlap among genes with opposite changes. We realized that the RRHO results were not explained in adequate detail. We have updated the manuscript (page 5) to clarify the interpretation.

There are differential editing sites between E and M tumors, and these differential editing sites affect the host mRNAs abundance. However, according to the data (Fig. 2B, 5D, and 6E), such effect is subtle. There is lack of evidence (experiments or publications) to show subtle alteration of these mRNAs results in EMT.

We understand this concern and agree with the reviewer that the statistically significant results do not always reveal large magnitude of differences. We appreciate the reviewer's recognition that there might exist a difference between conventional biological interpretations and global bioinformatic conclusions. As for studies on RNA editing, a large amount of effort in the past has been dedicated to understanding the function of a small number of editing sites. These editing sites were found to have interesting biological impacts, often large in magnitude. However,

recent sequencing-based studies revealed an unprecedented number of A-to-I editing sites in the human transcriptome, with the average editing level being 0.1-0.2. It remains largely unknown whether many of these editing sites are functional, and if yes, what types of function they may have. Given the large number of editing sites, bioinformatic approaches can be powerful in revealing global trends in the data and making novel discoveries.

However, since most of the endogenous A-to-I editing sites have low editing levels, the observable endogenous functional impacts will unfortunately be small on average. Despite the small amplitude, such impacts still reflect *bona fide* functions of RNA editing. It is also possible that the functions of some editing sites become more pronounced in specific contexts, such as during stress or in certain diseases. In the future, to better understand the biological relevance of a large number of editing sites each imposing a small change to gene expression, it is very likely that systems biology approaches are necessary. That is, the small change caused by RNA editing to one gene's expression may not be biologically significant, but the collective changes in many genes or pathways may very well make a significant collective biological impact. We are currently exploring this topic, which is outside the scope of this manuscript. We hope that these systems-level approaches can lead to future breakthroughs in our understanding of the biological relevance of RNA editing.

Fig. 4. Since both ADAR1 and ADAR2 are expressed in A549 and MCF10A cells, to rule out the possibility of compensation between two ADAR proteins, it is necessary to detect the editing level of some sites (differentially edited between E and M) after ADAR knockdown. Besides, the authors may need to perform rescue assays of ADAR loss experiments to solidify their finding.

As suggested, we carried out knockdown (KD) experiments of ADAR1, ADAR2 or both proteins (and control experiments) in A549 cells, followed by RNA-seq of three biological replicates each. The RNA editing profiles between each of the three ADAR KD conditions and control were compared (Fig. R4A). We observed a large number of editing sites with significant editing changes upon ADAR1 KD (detected by REDIT-LLR [1]). Comparing ADAR1 KD and double KD, very few editing sites were differential, suggesting that ADAR2 does not substantially compensate for loss of ADAR1 editing in A549 cells.

In contrast, a relatively small number of editing sites had significant editing changes upon ADAR2 KD (Fig. R4A). In addition, some editing sites had increased levels upon ADAR2 KD. Thus, it is likely that ADAR1 compensates for ADAR2 for a relatively small number of editing sites. Due to such compensation, the EMT phenotype observed upon ADAR2 KD may have been buffered by ADAR1 compensation. Yet, our experiment (Fig. 4) showed detectable EMT changes upon ADAR2 KD, suggesting that the changes induced by loss of ADAR2, despite compensation by ADAR1, are strong enough to induce the EMT phenotype. Thus, our observation (Fig. 4) holds despite the existence of compensation. ADAR1 and ADAR2 compensation is a fascinating topic. We appreciate this comment raised by the reviewer and plan to investigate further in our future work.

Next, we examined the differential editing sites between E and M LUAD and LUSC tumors to determine if they responded to single or double ADAR KD in the above data. We observed that the majority of these sites had decreased editing levels upon ADAR1 KD (but not ADAR2 KD) and a small fraction of sites responded to ADAR2 KD (but not ADAR1 KD). Sites that are regulated by both ADARs also comprise a small fraction of the total (Fig. R4B). Thus, we conclude that compensation between the two ADAR proteins is not a dominant factor for differential editing sites observed in LUAD and LUSC.

We have revised the manuscript to include the above results (Supplementary Fig. 6, page 7). Since a large number of editing sites are involved as ADAR targets, it is not practical to carry out rescue experiments that encompass so many sites. We would pursue such an experiment in the future if technologies allow.

Reference

1. Tran SS, Zhou Q, Xiao X. Statistical inference of differential RNA-editing sites from RNA-sequencing data by hierarchical modeling. *Bioinformatics*. 2020;36:2796–2804.

Figure R4. Altered editing upon knockdown of ADAR1, ADAR2, or both.

A. Mean editing levels of testable sites in five comparisons between ADAR KD conditions or

control experiment. Sites with significant editing differences between conditions (REDIT-LLR p-value < 0.05 and magnitude of mean editing difference ≥ 0.05) are colored red, while gray represents nondifferential sites. Y=x line shown in blue. **B.** Proportions of lung cancer E-M differential sites that were also differential in ADAR KD conditions (compared to controls). sigADAR1: sites that were differential only in ADAR1 KD. sigADAR2: sites that were differential only in ADAR2 KD. sigBoth: sites that were differential in both ADAR1 KD and ADAR2 KD, or in double KD. The prefix 'red' indicates reduced editing level by at least 0.05 upon KD from control, but did not pass the statistical significance requirement. 'Remain': editing sites that were not significantly different or reduced across any comparison.

Fig. 1B shows that editing level increases in M tumor compared to E in both LUAD and LUSC, suggesting that editing has a "pro-EMT" effect. However, Fig. 4 shows that the loss of ADAR promotes EMT in A549 cells, suggesting editing has an "anti-EMT" effect. The conflicting results need to be explained.

This is a great point. These observations reflect the complex nature of the potential mechanisms of RNA editing regulation and function in cancer. The fact that ADAR KD in cell culture caused EMT changes suggests that RNA editing changes may drive EMT or at least constitute an important aspect of EMT mechanisms. However, in the primary tumors, we did not observe a high level of difference of ADAR expression levels between E and M tumors that are consistent with the observed RNA editing differences for all cancer types (response to the first point above). Thus, ADAR expression may not fully account for editome difference between E and M tumors. This difference may be due to regulation by other proteins that directly or indirectly affect ADAR function. We wish to investigate this question in our future work. The seemingly opposite trend in E vs. M RNA editing differences between bulk tumors and cultured cell lines may reflect the complexity of cancer biology that is not effectively recapitulated by cell culture models. Although the cell culture models can support the likely importance of RNA editing in EMT, the exact mechanisms and related regulation can only be investigated using in vivo models in the future. We have now updated the Discussion (page 11) to clarify the above points.

Since ILF3 is required to regulate the abundance of mRNAs containing differentially edited sites, it would be very helpful to evaluate if ILF3 ablation affects EMT.

This is an interesting question. We did ILF3 knockdown (KD) via two different siRNAs in A549 cells and examined the EMT characteristics via cell morphology and gene expression. We observed that the cell morphology changed from tightly connected round cells towards the more dispersed, spindle-shaped cells (Figure R5A). We also detected a minor decrease in the epithelial marker E-cadherin and an increase in the mesenchymal marker N-cadherin in the ILF3 KD cells (Figure R5B). We further confirmed these changes via qRT-PCR (Figure R5C). Thus, these data support that ILF3 KD induced EMT, consistent with the expectations of our model. Nonetheless, it should be noted that the EMT changes upon ILF3 KD were not as pronounced as those observed upon ADAR KDs. This is perhaps not surprising because ILF3 is

known to have very diverse functions in gene regulation [1-2]. The various functional pathways affected by ILF3 KD in cancer EMT need detailed investigation in the future.

We have included Figure R5 in the new Figure 7C-E and revised the manuscript accordingly (page 10).

References:

- 1.Castella, Sandrine, et al. "Ilf3 and NF90 functions in RNA biology." Wiley Interdisciplinary Reviews: RNA 6.2 (2015): 243-256.
- 2.Freund, Emily C., et al. "Unbiased identification of trans regulators of ADAR and A-to-I RNA editing." Cell Reports 31.7 (2020): 107656.

Figure R5. ILF3 KD induced partial EMT in A549 cells. A. Images of A549 cells transfected with siRNAs targeting ILF3 (two different siRNAs were used to KD ILF3, siILF3_1 and siILF3_2) or control siRNAs (siControl). Scale bars: 100um. **B.** Western blot detecting protein levels of ILF3, E-cadherin, N-cadherin and internal control β -Actin in the siControl, siILF3_1 and siILF3_2 A549 cells. Three biological replicates were carried out for each experiment. **C.** Normalized mRNA expression levels (mean \pm SD) for ILF3, E-Cadherin and N-Cadherin in the siControl, siILF3_1 and siILF3_2 A549 cells. Three biological replicates were carried out for each experiment. The expression values were compared between siILF3 and siControl via t-test. **p<0.01, ***p<0.001, ****p<0.0001, n.s., not significant.

Additional experiments are required to check whether editing will affect ILF3 binding to its target mRNAs.

To address this question, we first examined the ILF3 eCLIP reads to identify those that encompass A or G at a specific editing site. If an editing site affects ILF3 binding, we expect a bias in the relative amount of A- or G-containing eCLIP reads. However, the A:G relative read ratio in eCLIP peaks needs to be compared to an expected A:G ratio that reflects the initial editing level before editing-dependent regulation takes place. In general, this expected A:G ratio is hard to estimate using regular RNA-seq data. To address this challenge, we used RNA-seq data of the nuclear fraction. Although not perfect estimates, the A:G ratios in nuclear RNA can be used as a proxy of initial editing levels before downstream regulation occurs. Figure R6A shows the comparison of editing levels observed in ILF3 eCLIP reads and those in nuclear RNA-seq data of A549 cells. There exists a trend of bias toward higher editing levels in ILF3 binding sites ($p = 0.06$, Proportion test), although not statistically significant possibly due to small sample size. This observation indicates that ILF3 may have stronger binding to the edited RNA than the unedited counterpart. Nonetheless, we do acknowledge that the number of editing sites included here is small, due to limited sensitivity of eCLIP in general. In addition, it is possible that an editing site may affect ILF3 binding without falling within the significant ILF3 eCLIP peaks, but in the vicinity of the peaks, given that ILF3 is a double-stranded RNA binding protein.

To examine this question more broadly, we also analyzed nuclear and cytosolic RNA-seq data of A549 to compare levels of editing sites in putative target genes whose expression is modulated by RNA editing. The assumption is that cytosolic editing levels are resulted from additional post-editing regulation that does not occur in the nuclear fraction. Thus, if editing-dependent regulation of mRNA stability exists, such as exerted by ILF3, we expect an overall bias between the cytosolic and nuclear editing levels. As shown in Fig. R6B, significantly more editing sites had higher editing levels in the cytosolic fraction than the nuclear fraction ($p = 0.0009$, Proportion test). This observation is consistent with a general model that RNA editing tends to stabilize the mRNA, such as by preferentially binding to ILF3.

Figure R6. Comparison of editing levels across mRNA subsets in A549 cells. A. Mean editing levels in ILF3 eCLIP-seq and nuclear RNA-seq for ADAR KD-supported expression-correlated sites (as described in the manuscript) that were located in significant ILF3 eCLIP peaks. Mean editing levels of replicates (two for eCLIP, two for nuclear RNA-seq) were

calculated for sites covered by at least 10 reads with at least 2 edited reads. The n values indicate the numbers of sites above and below the blue $y = x$ line. **B.** Similar as (A), but showing mean editing levels in cytosolic and nuclear RNA-seq data.

Alternative to bioinformatic data, EMSA-type of experiments are needed to provide more direct proof of editing-dependent ILF3 binding. However, the EMSA experiments need to include long double-stranded RNAs with structures mimicking their in vivo forms. This is very challenging since in vivo structure is very hard to reproduce during in vitro experiments.

Although the bioinformatic results support the hypothesis of editing-dependent ILF3 binding, we decided not to include these data cautioning the lack of direct experimental binding data. If the reviewer thinks the data are still valuable, we'd be happy to include it in a revised manuscript.

Minor concerns:

Fig. 2C. The authors may need to perform a GO analysis of all the edited mRNAs as a control. Besides, it would be better to perform the GO analysis on hyper- and hypo- edited gene lists separately.

The performed GO analysis used non-differentially edited mRNAs as controls to compare to differentially edited ones, so enriched terms represent those enriched relative to the background of non-differentially edited mRNAs. Since these controls were incorporated into the analysis, we did not repeat it on all edited mRNAs. As the reviewer recommended, we performed separate GO analyses on hyperedited and hypoedited genes in each cancer type. As seen in the figure below, GO enrichment among hyperedited genes is similar to that among all differentially edited genes, for cancer types with a global trend of hyperediting in M. Similarly, for hypoedited tumor types (BRCA and OV), the GO enrichment among hypoedited genes is similar to that of all differentially edited genes. We agree that the results on hyper- and hypoedited genes separately may be informative, and have included the figure as Supplementary Fig. 2.

Figure R7. Gene ontology enrichment among differentially edited genes. Significance of enrichment of gene ontology (GO) terms among all differentially edited genes (blue), only hyperedited genes (green) or only hypoedited genes (pink) of each cancer type. Point size represents the statistical significance of enrichment (\log_{10} -transformed adjusted p-value). Terms significantly enriched in at least two cancer types are shown.

Fig. 3A. The abbreviation "EC" is not explained - it may be endothelial cells.

Thank you. We have updated the text to introduce EC as the abbreviation of endothelial cells.

Page 6, Line 19-23. The difference in gene expression between different cell types is subtle (Fig. 3A). Thus, this result is not sufficient to conclude that "the editing differences observed among bulk tumors may be predominantly attributable to the cancer cells". The authors need to compare the editing in different cell types in the tumor.

Thank you for raising this point. In addition to examining gene expression differences among cell types, we have compared the extent of editing of sites that were differential between E and M tumors. Specifically, for each set of differential editing sites identified in bulk tumors of a

cancer type, we calculated the proportion of differential sites that were edited in each cell type of the lung cancer scRNA-seq data. As shown in the figure below, the highest proportion of edited sites was observed in cancer cells, and this proportion was significantly greater than those of other cell types. The highest prevalence of editing at differential sites in cancer cells supports our conclusion that the bulk tumor editing differences are most likely occurring in cancer cells.

We have updated the manuscript to include the above result (Figure 3B, page 6).

Figure R8. Proportion of differential editing sites from bulk tumor analysis that were edited in individual cell types of the lung cancer scRNA-seq data. A site was considered as edited in a cell type if the site was covered by at least 5 reads and the editing event was supported by at least 2 reads. Each point represents the proportion of sites from one cancer type. The proportions for the two top cell types were compared by Mann Whitney U test, with significance of p-value ** indicating $p \leq 0.01$.

Fig. 3C. I only saw 8 sig points in figure (9 described in the context). The difference seems subtle to me.

Thank you for pointing out the apparent discrepancy in the number of differential editing sites, which was due to one significant point plotted underneath a nonsignificant one. We have updated the figure so that all significant points are in the forefront of the plot. Although all 9 sites passed the statistical significance test, some of them had relatively small differences. Nevertheless, the shared direction of editing differences among all nine significant sites, consistent with the global hyperediting trend in bulk tumors of the same cancer type (LUAD), may support a genuine distinction in editing profiles between E and M phenotypes in tumor cells.

Fig. 6A. It would be better to use unchanged editing sites in the same mRNAs as control rather than randomly picked adenosine.

Thank you for this suggestion. We performed a modified eCLIP peak distance analysis, using nondifferential editing sites within the same mRNAs as controls. Given that much fewer

nondifferential sites than adenosines were present in each gene, we had to use fewer (100) sets of random controls to test the significance of the AUC of differential editing sites. The distance between ILF3 peaks and differential editing sites is again significantly close compared to the control editing sites in HepG2 and A549 cells, but this difference was not significant in K562 cells (Figure below). We caution that using nondifferential editing sites did not allow an adequate number of distinct random controls and some nondifferential sites may be *bona fide* ILF3 relevant editing sites but failed the statistical test due to inadequate power. Thus, we chose to use the random adenosines as the controls in this analysis.

Figure R9. Histograms of distances from differential editing sites to the closest ILF3 eCLIP peaks in editing-correlated genes in three cell lines (A549, K562, and HepG2), up to 10 kb (turquoise). For each cell line, gray curves represent distances for 10 sets of controls (nondifferential editing sites in the same genes as differential editing sites). Number of differential editing sites is given by n. P-value was calculated by comparing the area under the curve (AUC) of the distance distribution for differential editing sites to a normal distribution fit to the AUC values of 100 sets of random gene-matched nondifferential editing sites.

Second round of review

Reviewer 1

I appreciate the authors detailed response and additional data added to the manuscript.